# Therapeutic Potential of HLA-I Polyreactive mAbs Mimicking the HLA-I Polyreactivity and Immunoregulatory Functions of IVIg

**DOI:** 10.3390/vaccines9060680

**Published:** 2021-06-21

**Authors:** Mepur H. Ravindranath, Fatiha El Hilali, Edward J. Filippone

**Affiliations:** 1Department of Hematology and Oncology, Children’s Hospital, Los Angeles, CA 90027, USA; 2Emeritus Research Scientist at Terasaki Foundation Laboratory, Santa Monica, CA 90064, USA; 3Mohamed V. Hospital, Mekines 50060, Morocco; hfatiha@gmail.com; 4Division of Nephrology, Department of Medicine, Sidney Kimmel Medical College at Thomas Jefferson Univsity, Philadelphia, PA 19145, USA; kidneys@comcast.net

**Keywords:** intravenous immunoglobulin (IVIg), human leukocyte antigen-I (HLA-1), polyreactive mAbs, monospecific mAbs, shared epitopes, immunosuppression, T-cells, B-memory cells, T-regulatory cells, blastogenesis, proliferation, antibody production

## Abstract

HLA class-I (HLA-I) polyreactive monoclonal antibodies (mAbs) reacting to all HLA-I alleles were developed by immunizing mice with HLA-E monomeric, α-heavy chain (αHC) open conformers (OCs). Two mAbs (TFL-006 and TFL-007) were bound to the αHC’s coated on a solid matrix. The binding was inhibited by the peptide ^117^AYDGKDY^123^, present in all alleles of the six HLA-I isoforms but masked by β2-microglobulin (β2-m) in intact HLA-I trimers (closed conformers, CCs). IVIg preparations administered to lower anti-HLA Abs in pre-and post-transplant patients have also shown HLA-I polyreactivity. We hypothesized that the mAbs that mimic IVIg HLA-I polyreactivity might also possess the immunomodulatory capabilities of IVIg. We tested the relative binding affinities of the mAbs and IVIg for both OCs and CCs and compared their effects on (a) the phytohemagglutinin (PHA)-activation T-cells; (b) the production of anti-HLA-II antibody (Ab) by B-memory cells and anti-HLA-I Ab by immortalized B-cells; and (c) the upregulation of CD4+, CD25+, and Fox P^3^+ T-regs. The mAbs bound only to OC, whereas IVIg bound to both CC and OC. The mAbs suppressed blastogenesis and proliferation of PHA-activated T-cells and anti-HLA Ab production by B-cells and expanded T-regs better than IVIg. We conclude that a humanized version of the TFL-mAbs could be an ideal, therapeutic IVIg-mimetic.

## 1. Introduction

Personalized, passive immunotherapy encompasses the transfer of monoclonal antibodies (mAbs), polyclonal antibodies (Abs), and purified natural Abs for the specific purpose of either upregulating or suppressing immune functions, depending on the nature and status of the underlying disease. A variety of mAbs, capable of performing specific functions, are available. These include antibodies capable of depleting various cell types such as B-cells (rituximab), CD52-bearing leukocytes (alemtuzumab), and CD33-bearing cells (gemtuzumab). Additionally, mAbs can inhibit various cytokine pathways such as the TNF–TNFR axis (etanercept, adalimumab, infliximab) and the IL-6-IL-6R axis (tocilizumab, actemra). In contrast, therapeutic intravenous immunoglobulin (IVIg) contains purified natural Abs, prepared by pooling and purifying IgG from the plasma of 10 to 60 thousand normal and healthy donors. The United States Food and Drug Administration (FDA) has approved immunoprophylaxis with IVIg therapy for chronic inflammatory demyelinating polyneuropathy (CIDP), immune thrombocytopenic purpura (ITP), primary immunodeficiency (PI), secondary immunodeficiency in chronic lymphocytic leukemia, pediatric human immunodeficiency virus (HIV) infection, Kawasaki disease, prevention of graft vs. host disease in adult bone marrow transplant recipients, and organ transplantation [1]. 

The presence of pre-existing Abs in a potential organ transplant recipient is referred to as sensitization. It remains one of the significant immunological barriers to receiving an organ. As well as restricting the availability of a given organ for a particular patient, these antibodies cause long-term graft attrition when present post-transplantation [2,3]. IVIg is administered to end-stage organ disease patients pre-transplantation and allograft recipients post-transplantation for reduction (desensitization) of these HLA Abs. The exact mechanism of desensitization by IVIg has not been elucidated. However, IVIg *per se* contains polyreactive, natural IgGs against multiple targets, including endogenous and exogenous Abs, immunomodulating peptides, blood group antigens, and various cytokines. Whereas some of the immunoregulatory mechanisms of the action of IVIg have been proven in vitro and animal models, many proposed mechanisms remain enigmatic due to IVIg polyantigen reactivity, polyclonality, and diversity in the preparations of IVIg. Several mechanisms of action of IVIg have been proposed [4]. These include (a) Fc-receptor blockade, (b) neutralization of Abs by idiotypic and anti-idiotypic Abs, (c) blockade of the Fas apoptotic pathway by anti-Fas auto-Abs, (d) regulation of complement components, (e) modulation of cytokine secretion, (f) hindrance of natural-killer cell activity, (g) inhibition of matrix metalloproteinase-9, (h) suppression of NF-kB activation and IkB degradation, (i) G1 cell cycle arrest, (j) prevention of tumor growth, (k) decrease in leukocyte recruitment, (l) attenuation of T-cell stimulation, (m) effects on Ab kinetics, and (n) effects on dendritic cells. These mechanisms of IVIg are not mutually exclusive and operate synergistically. 

Additionally, more observations have accrued to document that normal, healthy human sera contain a diverse variety of IgG Abs against allo-HLA molecules [5,6,7,8,9]. Therefore, the IVIg purified from the pooled plasma of thousands of healthy individuals clearly carries allo-HLA Abs.

Our research commenced with studying the structure of a non-classical HLA class-I (HLA-I) antigen, namely HLA-E and anti-HLA-E mAbs [10,11]. Anti-HLA antibodies do not react with an entire antigen; like all antibodies, the reaction is against a portion of the antigen termed an epitope. At the center of the epitope is an eplet consisting of one to several amino acids recognized by the complementarity-determining regions of the antibody. Epitopes may be specific to a single HLA locus or HLA allele (private epitopes, vide infra). Other times, the epitope may be shared by other loci or alleles (public epitopes, vide infra). The unique and shared amino acid sequences of the diverse HLA-I antigens were identified to elucidate whether the mAbs generated by immunizing an HLA-I antigen recognized public or private epitopes. In addition, the relative immunoregulatory capability of antibodies against either public or private epitopes deserves consideration. The objective of this review is to compare the nature and functional characteristics of the polyreactive anti-HLA-E mAbs (TFL-006 and TFL-007) with those of the commercial preparations of therapeutic IVIg as follows: Determine if HLA-E shares antigenic amino acid sequences (epitopes) common to all other HLA-I isoforms;Document HLA-I polyreactivity of HLA-E mAbs;Document HLA-I polyreactivity of the therapeutic preparations of IVIg;Compare immunomodulation by IVIg with polyreactive mAbs
○Suppression of T-cell proliferation;○Suppression of antibody production by B cells;○Expansion of Foxp^3^+ Tregs. 


## 2. HLA-E Shares Antigenic Amino Acid Sequences (Epitopes) Common to All Other HLA-I Isoforms

HLA antigens, located on the surface of all nucleated cells of the human body, are of two different major classes, HLA-I and HLA class-II (HLA-II). The genes that encode HLA-I and HLA-II are closely associated with each other on the short arm of human chromosome 6. The entire complex of HLA encompasses four million base pairs of DNA and “is of a size comparable to the genome of *Escherichia coli*” [12].

The HLA-I molecule is a heterodimer with a 45 kDa α-heavy chain (αHC) anchored to the bilayer lipid membrane. It is complexed with a water-soluble, 12 kDa light chain called β2microglobulin (β2m) (Figure 1), encoded by a gene located on chromosome 15. The HLA-I loci consist of the classical HLA-Ia and non-classical HLA-Ib loci, each containing three loci: HLA-Ia (HLA-A, HLA-B, and HLA-C) and HLA-Ib (HLA-E, HLA-F, and HLA-G). The α C of HLA-I consists of different domains encoded by different exons. The leader peptide is encoded by exon 1, and the three extracellular domains (α1, α2, and α3) are encoded by exons 2, 3, and 4 respectively. The transmembrane domain is encoded by exon 5, and the cytoplasmic tail by exons 6 and 7. The antigenic polymorphism of different isoforms is reflected in the amino acid sequences. As noted above, the sequences or epitopes characteristic of each allele of each locus are referred to as its “*private epitopes*” or “specific sequences.” However, all isoforms also share several common or shared sequences with other alleles of the same locus or other loci, referred to as “*public epitopes*”. 

### 2.1. Private and Public Epitopes of HLA-E

Table 1 shows both HLA-E amino acid sequences that are specific and those that are shared with alleles of other loci of HLA-I. Some amino acid sequences or epitopes are specific to the HLA-E locus because they are not found in other HLA-Ia or Ib loci. The most striking private epitopes or sequences are ^65^RSARDTA^70^ and ^143^SEGKSNDASE^152^. Table 2 summarizes the results of comparing HLA-E sequences with hundreds of alleles from the HLA-Ia and HLA-Ib loci. Some sequences of HLA-E are found in only one allele. The HLA-E sequences, or PRAPWMEQE, and EPPKTHVT are found in HLA-A*3306 and HLA-B*8201, respectively, whereas HLA-E sequences ^117^AYDGKDY^123^ and ^126^LNEDLRSWTA^135^ are shared with all other HLA loci (HLA-A, HLA-B, HLA-C, HLA-F, and HLA-G) that we have examined. These latter sequences perfectly befit the definition of “shared” or “public epitopes.”

### 2.2. The Most Widely Shared Public Epitopes Are Cryptic in HLA-I

The HLA-I molecule is a heterodimer because the α-HC is closely associated with β2m (Figure 1). As a consequence, not all sequences of HLA α-HC are exposed. Figure 1A shows that the specific sequences, or the private epitopes, of HLA-E (shown in yellow), ^65^RSARDTA^70^ and ^143^SEGKSNDASE^152^, are exposed for direct binding of Abs. Figure 1B shows that the most commonly shared sequences or the public epitopes (shown in yellow) ^117^AYDGKDY^123^ and ^126^LNEDLRSWTA^135^ are cryptic, as they lie in close proximity with β2m. Figure 2 illustrates the crypticity of the shared epitope, ^117^AYDGKDY^123^. The exposure of these shared epitopes depends on the nature of the HLA molecule. Most often, the HLA molecules occur as heterodimers, also designated as “closed conformers” or CCs. In CCs, the most prevalent shared epitopes, ^117^AYDGKDY^123^ and ^126^LNEDLRSWTA^135^, are masked by β2m. However, in β2m-free α-HCs, designated as “open conformers” (OCs), they are exposed.

Thus, the HLA-I molecules occur as heterodimers complexed with β2m on the cell surface and as β2m-free α-HC OCs. Schnabl et al. [13] reported that T lymphocytes activated in vitro or in vivo, but not resting, expressed many OCs. Using the mAb W6/32 that specifically recognized HLA-I CCs and the mAbs L45 and HC10 that bound explicitly to OCs but not to CCs, they confirmed the existence of HLA-OCs. Immunoprecipitation and co-capping experiments showed that LA45 was bound to HLA-I OCs at the cell surface. The mAb L45 bound to PHA-activated T-cells from a panel of 12 people with different HLA types, suggesting that LA45 might bind to epitopes shared by all HLA-I α-HCs. The HLA-I OCs expressed on the cell surface of activated T-cells—or EBV-transformed B-cells—are referred to as “peptide-binding empty HLA” [14,15]. The presence of HLA-I OCs was confirmed on activated T-cells in vitro and in vivo and on B cell lines (RAJI, NALM6), EBV-transformed B-cells, and the myeloid cell line KG-1A [16,17]. Interestingly, the expression of OCs on the cell surface in normal human T-cells upon activation and cell division correlated with the level of proliferation [18]. Indeed, the OCs on the cell surface were glycosylated. The inhibition of tyrosine phosphorylation with the Src-family tyrosine kinase inhibitor PP2 resulted in the enhanced release of HLA-I α-HC from the cell surface of activated T-cells. Further studies indicated that the OCs were regulators of ligand–receptor interactions and had potential implications for immune activation [18,19,20] and immune regulation [21]. The inflamed tissues in spondyloarthropathy showed increased levels of OCs on CD14b+ monocytes compared to other leukocyte subsets. The level of OCs also increased on activated dendritic cells of the extravillous trophoblast [22]. Interestingly, the OCs maintained the peptide-binding groove in vitro [23]. Khare et al. [24] reported that the β2m-free HCs of HLA-B27 might induce arthritis in transgenic mice, and β2m-free, HC-specific Abs decreased the disease incidence in this model. OCs exist as dimers or in clusters at the cell surface in vivo [23,25,26], profoundly impacting receptor engagement. Recognition of OCs as ligands by members of the killer Ig receptor family (KIR) and Ig-like transcript (ILT)/LIR/LILR family could influence their immunomodulatory function in inflammatory disease [27]. The shared sequences presented in HLA-E OCs contribute to the generation of HLA-I polyreactivity Abs.

Though the OCs are found on activated immune cells, the soluble forms of HLA-I present in humans also occur as OCs. Demaria et al. [28] found that the levels of OCs in circulation were regulated by proteolytic cleavage. The release was mediated by a Zn^(2+)^-dependent, membrane-bound metalloprotease (MMP). Specific separation by the metalloprotease occurs at a site close to the papain cleavage site in the α3 domain of HLA-I HCs. This site is not accessible to MMP in CCs. During the shedding of HLA-I trimers under different pathological conditions, the exogenous peptide and β2m dissociate from these CCs, and subsequent cleavage of OCs (Figure 3) may be partially responsible for controlling the levels of OC on the surface of activated cells. Since the shed OCs expose previously cryptic epitopes, they may be recognized by B-cells for antibody production.

### 2.3. Antigenicity Rank of the Most Widely Shared Public Epitopes of HLA-E

An Ab directed against a protein antigen can recognize either a linear or a discontinuous sequence (still closely aligned in space due to protein folding) in its native conformation. This so-called eplet is present at the center of an epitope. The immunogenicity of an epitope is defined as its ability to actively induce an immune response. In contrast, the antigenicity of an epitope is defined as its passive ability to be recognized by the immune system. The antigenicity of a sequence, or epitope, in a polypeptide can be predicted using parameters such as hydrophilicity, flexibility, accessibility, beta turns, cell surface exposure, and polarity. Several methods are known for predicting the antigenicity of a continuous sequence. Chou and Fasman [31] developed a methodology, based on α-turns in protein secondary structures, for predicting antigenicity. Kolaskar and Tongaokar [32] developed a semi-empirical method using the physicochemical properties of amino acid residues and their frequency of occurrence in segmental epitopes. Karplus and Scholtz [33] predicted antigenicity based on three flexibility scales. Parker et al. [34] predicted antigenicity with a novel hydrophilicity scale. In these methods, when computing the score for given residue *i*, the amino acids in an interval of the chosen length, centered around residue *i*, are considered.

Further details are provided on the antibody epitope prediction web (http://tools.immuneepitope.org/tools/bcell/iedb_input (Last updated on 30 May 2021, see also iedb.org for details). The methods were collectively employed to assess the antigenicity ranking of the different sequences of HLA-E listed in Table 2**.** Among the shared peptide sequences listed in the table, the most prevalent shared epitope, ^117^AYDGKDY^123^, ranked first in antigenicity. We have used these peptide sequences and the second-ranking epitope for peptide inhibition studies of anti-HLA-E mAbs and established that mAbs TFL-006 and TFL-007 are the most polyreactive [10,11,35,36].

## 3. Documentation of HLA-I Polyreactivity of HLA-E mAbs

### 3.1. Methodology–Development and Characterization of HLA-E mAbs

#### 3.1.1. Anti-HLA-E mAb Production

The mAbs were produced following the recommendations of the National Research Council’s Committee on Methods of Producing mAbs [37]. The recombinant α-HCs HLA-E^R107^ and HLA-E^G107^ (source: Immune Monitoring Laboratory, Fred Hutchinson Cancer Research Center, concentration: 10 mg/mL in MES buffer) were used. Two mice were immunized with 50 mg of each antigen in PBS (pH 7.4, 100 mL) admixed with TiterMax (100 mL) (Sigma-Aldrich, St Louis, MO, USA) adjuvant before injection into the footpad and intraperitoneally. Three immunizations were given at about 12-day intervals, with an additional immunization after 12 days for mice receiving HLA-E^G107^. 

The clones were cultured in a medium (RPMI 1640 w/L-glutamine and sodium bicarbonate, with 15% FBS, penicillin, streptomycin, L-glutamine, and sodium pyruvate. Hybridomas were cryopreserved in RPMI containing 10% DMSO and 20% FBS. All culture supernatants were screened for IgG reacting to HLA-A, HLA-B, HLA-C, HLA-E, HLA-F, and HLA-G using single-antigen microbeads on a Luminex platform. The mean fluorescent intensity (MFI) was determined for supernatants after diluting to 1/2. The MFI values were corrected against those obtained with negative control beads. Further details are provided elsewhere [35,38].

#### 3.1.2. HLA-1 Antigen-Coated Single Antigen Beads (SABs) and the Luminex Platform for Monitoring the Reactivity and the Density of These mAbs

The affinity of Abs to different HLA-I antigens was monitored on a Luminex Platform, using HLA-I molecules, coated as single antigens, on the multiplex, fluorescinated microbeads. Three different kinds of microbeads were available for investigation, as illustrated in Figure 4: (i) beads with an admixture of CCs and OCs (Regular LABScreen SAB, source: One Lambda, Inc., Canoga Park, CA, USA), (ii) beads restricted to CCs only (iBeads developed by trypsinization of regular LABScreen beads and Immucore LIFECODES SAB), and (iii) beads restricted to OCs (alkali or acid-denatured beads). The regular LABScreen microbeads were coated with HLA-A (31 different antigens), HLA-B (50 different antigens), and HLA-C (16 different antigens). In addition, the HLA-Ia microbeads had built-in control beads: positive beads coated with human IgG and negative beads coated with serum albumin (human or bovine). For HLA-Ib, the control beads (both positive and negative) were added separately. 

The different kinds of microbeads were characterized using three different mAbs (Figure 4). The mAb W6/32 (IgG2a) bound to CCs, but not to OCs [39], and bound to both peptide-associated and peptide-free CCs [40,41]. The mAb HC-10 (IgG2a) recognized an epitope in HLA-I between amino acid positions 57 and 62, with arginine at position 62 (R62) being crucial for HC-10 binding [42]. HC-10 recognized cell surface CCs devoid of a peptide, whereas the presence of a peptide reduced HC-10 reactivity [43]. The mAb TFL-006 (IgG2a) bound to OCs of all HLA-I loci and was inhibited by peptides from the amino acid sequences shared by all HLA-I loci [35,36].

### 3.2. The HLA Reactivity Groups of mAbs Generated by Recombinant HLA-E OCs

More than 200 hybridomas were generated using the OCs of HLA-E. All mice experiments were carried out under the guidance of Professor Paul I. Terasaki, who owned One Lambda, Inc., in Canoga Park, CA, USA, with animal subject committee approval. The mAbs secreted by these hybridomas were both HLA-E monospecific and HLA-Ia and HLA-Ib polyreactive. These mAbs could be categorized into eight different groups, as shown in Table 3. Group 1 corresponds to monospecific mAbs reacting restrictively with HLA-E. Group 4 refers to HLA-Ib-specific mAbs. Group 10 represents mAbs, recognizing both HLA-Ia and HLA-Ib molecules (Table 4). 

Table 4 compares the HLA-I reactivities of an HLA-E monospecific mAb, TFL-033, with two HLA-I polyreactive mAbs, namely TFL-006 and TFL-007. TFL-006 showed reactivity with 32 alleles of HLA-A, 48 alleles of HLA-B, and 16 alleles of HLA-C, whereas TFL-007 reacted with 24 HLA-A, 44 HLA-B, and 16 HLA-C alleles. TFL-006 did not react with B*4901 and B*5701, although shared sequences (public epitopes) were present. TFL-006 and TFL-007 reacted with all three HLA-Ib isoforms as follows: HLA-E (+++) > HLA-G (++) > HLA-F (+). Among HLA-Ia isoforms, the reactivities in general were as follows: HLA-C (+++) > HLA-B (++) > HLA-A (+). The MFIs of the mAbs against A*1101 were the highest among HLA-A isoforms, with B*4006 the highest among HLA-B isoforms and C*1802 the highest among HLA-C isoforms. Of the seven HLA polyreactive mAbs belonging to group 10, TFL-006 ranked first, and TFL-007 ranked second. The reactivities of other mAbs of group 10 are presented elsewhere [44]. 

To determine the affinity of the mAbs TFL-006 and TFL-007 for shared sequences in the OCs, synthetic peptides purified by reverse-phase HPLC from GenScript Corporation (Piscataway, NJ, USA) were obtained [10]. Table 2 compares peptide inhibition using three commonly shared peptides, namely DTAAQI, AYDGKDY, and LNEDLRSWTA. These peptides were used, separately, to block the binding of the mAbs to the regular LABSCreen SABs, in which the OCs (β2m-free α-HC) were admixed with intact HLA trimers (CCs). Of the three peptides tested, AYDGKDY blocked more than 52% of the binding to the mAb TFL-006 compared to the other two peptides. Similar results were obtained with mAb TFL-007. 

To further ascertain that the mAbs recognized only the OCs, we compared the binding of mAb TFL-006 on LABScreen SABs (One Lambda, Inc., Canoga Park, CA, USA), which contained an admixture of OCs and CCs and with LIFECODES SABs (LSA Class I 03203F beads; Immucor, Norcross, GA, USA) that was devoid of HLA-I OCs [45,46,47,48]. The results presented in Table 5 show that the mAb TFL-006 did not bind to any of the alleles of the isoforms of HLA-I on LIFECODES SABs, confirming that mAb TFL-006 recognized only the OCs and did not bind to CCs on the LIFECODES beads. Earlier, we have shown that mAb TFL-006 binds well on acid-denatured LABScreen beads but not on iBeads, in which β2m-free HCs (OCs) are selectively, enzymatically removed from the regular LABScreen beads [45,46]. 

## 4. Documentation of HLA-I Reactivity of the Therapeutic Preparations of IVIg

The FDA, in 2004, approved the Cedars–Sinai IVIg desensitization protocol for minimizing allo-HLA Abs in patients waiting for kidney transplantation, given the known ability of HLA antibodies to destroy an allograft. Removal, or significant reduction, of such Abs, would then allow transplantations to proceed that would have otherwise been contraindicated. Since then, IVIg has emerged as a potential treatment strategy for desensitization protocols, pre-transplantation, and is now used post-transplantation for treating Ab-mediated rejection (AMR) caused by donor-specific anti-HLA Abs (DSA). Several clinical transplant centers adopted this strategy [2,3,49,50,51,52,53]. However, subsequent studies documented that IVIg preparations were often unable to reduce HLA Abs in the sera of transplant patients [54,55,56]. We examined five different therapeutic preparations of IVIg, namely GamaSTAN S/D (Talecris Biotherapeutics, Inc., Research Triangle Park, NC, USA), Sandoglobulin (6 gr, lot 4305800026; CSL Behring, Kankakee, IL, USA), Octagam (6 gr, lot A913A8431; Octapharma Pharmazeutika, Lachen, Switzerland); IVIGlob EX (VHB Life Sciences Limited, Bangalore, India) [34,35], and Immunoglobulin Normale (IV-LFB-CNTs LFB Biomedicaments, Courtaboeuf Cedex, France)[57]. 

HLA-I reactivities of the different dilutions (1/2 to 1/128) of IVIg preparations were tested on regular LABScreen SABs, acid-denatured LABScreen SABs (OCs only), and on enzymatically treated regular LABScreen SABs, called iBeads (CCs only). The details of this methodology are presented elsewhere [34]. Figure 5 compares the MFIs, signifying the density of HLA-Ia and HLA-Ib IgG Abs on the three different SABs for IVIg preparations from GamaSTAN (Figure 5A) and octagam (Figure 5B). Table 6 summarizes the HLA-I polyreactivity of commercial preparations of IVIg from different sources. The data presented in Figure 5A,B reveal that the strength of the Abs is much higher in denatured SABs, which is predominant with OCs, than in the CCs restricted to the iBeads. These findings confirmed that the anti-HLA IgG Abs in the IVIg preparations recognized both the CCs and OCs of HLA molecules, but with a higher prevalence of Abs recognizing OCs. To ascertain whether the immunoreactivity of IVIg to HLA-Ia alleles was due to the OCs of HLA-I, the anti-HLA-1a Abs were adsorbed out with Sephadex gel conjugated with OCs of HLA-E and then tested for HLA-E and HLA-Ia reactivity [35]. IVIg immunoreactivity to HLA-Ia was minimized after the adsorbing-out process. Most importantly, HLA-E OCs alone could absorb out a major fraction of anti-HLA-1a antibodies, indicating the prevalence of polyreactive antibodies against HLA-I OCs in IVIg preparations. This unique finding formed the basis for comparing the functional characteristics of IVIg with HLA-I polyreactive mAbs (TFL-006 and TFL-007).

## 5. Immunomodulation By IVIg Compared with Polyreactive Monoclonal Antibodies

### 5.1. Suppression of T-Cell Proliferation: IVIg vs. HLA-I-Polyreactive mAbs

#### 5.1.1. Background and Hypothesis

Activation of T-cells involves both blastogenesis and proliferation. Activation can be accomplished by natural or recombinant cytokines and PHA (phytohemagglutinin). Activation induces transitory expression of several molecules, both within the T-cell or on the cell surface. They include the IL-2 receptor, Fc receptors for IgG, insulin receptors, α fetoprotein, and transferrin receptors, MICA, HLA-II, and OCs of HLA-I [58]. Several studies documented that commercial IVIg inhibited PHA- or cytokine-induced T-cell activation and proliferation, both in vitro and in vivo [59,60,61,62,63,64,65,66,67,68,69,70,71,72,73]. 

Kaveri et al. [59] demonstrated that Abs to a conserved region of HLA-Ia, present in pooled therapeutic IVIg, were capable of modulating CD8+ T-cell-mediated function. Klaessn et al. [60] showed that IgG and F(ab)2 fractions in IVIg were responsible for the inhibitory function. In contrast, Miyagi et al. [61] attributed the inhibition of blastogenesis and proliferation of activated T-cells by IVIg to the Fc receptors for IgG, FcγRI (CD23), FcγRII (CD32), FcγRIII (CD16), and FcγRIV (CD64), expressed on the immune cells upon activation. MacMillan et al. [70] documented further the IVIg-mediated suppression of T-cell proliferation, with or without CD28 co-stimulation. Sali et al. [62] demonstrated that 2% intact or Fab_2_ fragments of IVIg could penetrate immune cells to modulate these activities. The penetrating fraction of IVIg inhibited the upregulation of activation marker CD25 on CD4+ splenocytes. In a placebo-controlled trial, the administration of IVIg to patients with inflammatory myopathies was associated with a significant reduction of the number of T-lymphocytes in vivo [74]. 

Based on these observations, it was hypothesized that HLA-I polyreactive mAbs that mimicked HLA-I reactivity of IVIg preparations might suppress activated T-cells, similarly to IVIg. The hypothesis was tested by comparing dose-dependent effects on the suppression of PHA-activated T-cells of three entities: IVIg, HLA-I polyreactive mAbs that mimicked IVIg (TFL-006 and TFL-007), and mAbs that did not mimic IVIg (TFL-033 and TFL-037). 

#### 5.1.2. Hypothesis Testing: Measurement of T-Lymphocyte Proliferation

More details are presented elsewhere, but all experimental analyses were carried out at Terasaki Foundation Laboratory with Review Board approval [58]. Briefly, we labeled purified human T-lymphocytes (freshly collected from a healthy, young adult male) with the intracellular fluorescent dye Carboxyfluorescein N-succinimidyl ester (CFSE), a cell-permeable dye that remained in the cell for several mitotic divisions. The technology used is illustrated in Figure 6A. Most importantly, the cessation of the progress of mitotic activity could be monitored as successive two-fold reductions in the fluorescent intensity after 72 h of treatment, including the addition of PHA and PHA plus IVIg or mAbs [58].

#### 5.1.3. The Suppression of Activated T-Cells by IVIg vs. mAbs

A summary of the findings is presented in Figure 6B, Figure 7 and Figure 8. The results established the differential effects on the suppression of PHA-activated T-cells, both by IVIg and by HLA-I polyreactive mAbs (TFL-006 and TFL-007). The mAbs (TFL-033 and TFL-037) that did not mimic the HLA-I reactivity of IVIg did not affect the activation or proliferation of T-cells. The mAbs TFL-006 and TFL-007 appeared to be more potent suppressors of the blastogenesis and proliferation of activated CD4+ T lymphocytes than IVIg. The concentrations of the mAbs required for the suppression of T-cell proliferation were 50-fold lower than the required concentration of IVIg. The suppression of blastogenesis and proliferation of T-cells by both IVIg and the anti-HLA-E mAbs was dose-dependent, and the dose required with mAbs was 50–150-fold lower than with IVIg. The mAb binding to OCs might have signaled T cell deactivation because the OCs have an elongated cytoplasmic tail with phosphorylation sites (tryosine320/serine335). 

A tentative model of PHA-mediated activation and HLA polyreactive, mAb-mediated deactivation is proposed (Figure 9). It is known that PHA activation initiates phosphorylation of the cytoplasmic domain of CD3 and the activation of transcription factors. TCR cross-linking leads to the expression and phosphorylation of cell surface molecules such as IL-Rα [74] and the OCs of HLA-I [13,14,15,16,17,18,19,20,21,22]. The binding of mAbs to the shared amino acid sequences or epitopes exposed on the OCs may dephosphorylate the tyrosyl and seryl residues on the elongated cytoplasmic tails of the HLA-I OCs [75,76]. This may simultaneously lead to dephosphorylation of CD3 and revert the PHA-activation of CD3 on T-cells (for further details, see the legend for Figure 9). Upon dephosphorylation, T-cells were deactivated, resulting in the suppression of blastogenesis and proliferation. Based on the observations that there was a suppression of the blastogenesis and proliferation of PHA-activated CD4+ T-cells by HLA-I polyreactive, anti-HLA-E mAbs (TFL-006 and TFL-007) but not by HLA-I non-reactive, anti-HLA-E mAbs (TFL-033 and TFL-007), it was inferred that the IVIg-mediated suppression of the blastogenesis and proliferation of PHA-activated CD4+ T-cells could also be due to binding of the HLA-I OC-reactive IgG fraction in IVIg. Furthermore, when equal concentrations of the HLA-I polyreactive mAb TFL-007 and IVIg were compared, the suppression by the mAbs was greater than that of the IVIg. Evidently, the admixture of other IgGs with HLA-polyreactive IgGs in IVIg might have hindered the T-cell suppressive efficacy of the IVIg. 

### 5.2. Suppression of Antibody Production by B-Cells: IVIg vs. HLA-I Polyreactive mAbs

#### 5.2.1. Background and Hypothesis

Previous literature documented that commercial IVIg not only inhibited PHA- and cytokine-induced T-cell activation and proliferation and significantly reduced the number of T-lymphocytes in vivo in a placebo-controlled trial in patients with inflammatory myopathies, but it was also capable of suppressing antibody production in patients under different disease conditions. Therefore, IVIg has become not only a substitution therapy for patients with immunodeficiencies [78], but also a therapeutic agent in autoimmune and systemic inflammatory diseases [79], as well as in organ and bone marrow transplantation [3,80].

IVIg is extensively used in patients with end-stage organ disease as well as allograft recipients. High levels of HLA antibodies, caused by various sensitizing events such as previous transplantation, pregnancy, or blood transfusion, are observed in patients with end-stage organ disease. High levels of such allo-HLA Abs can more likely produce positive crossmatch results with potential organ donors and preclude transplantation. Consequently, antibody-positive patients may experience prolonged waiting time. Among renal transplant recipients alone, such sensitized patients constitute more than one-third of those on the waiting list. For highly sensitized patients (with a panel-reactive HLA antibody (PRA) greater than 80%), the prospects of transplantation are grim. 

The formation of Abs against allo-antigens depends on both T and B-cells. Therefore, aggressive suppressive strategies have been developed to deplete both T and B-cells in order to reduce the generation of allo-HLA Abs formed before and after transplantation. One such strategy is induction therapy with rabbit or horse anti-human thymocyte globulin, a polyreactive, polyclonal mixture of non-specific cytotoxic Abs capable of killing immune cells [81]. However, many clinical transplant centers worldwide have formulated alternate protocols to suppress the formation of anti-HLA Abs. These protocols may include plasmapheresis (PP), high-dose IVIg, or a combination of PP with low-dose IVIg [2,49,50,51,52,53,54,82,83,84,85,86,87] or Rituximab, a monoclonal Ab (mAb) that depletes CD20+ B-cells [88,89,90,91]. Although several immunotherapeutic potentials are attributed to IVIg, its mechanism of action is far from certain. Possibly, it is due to polyclonality and the mixture of several kinds of IgG Abs present, including Abs against all HLA class I loci and alleles, as illustrated in Table 6, and Figure 5A,B.

A most interesting finding [35] was that the HLA-Ia reactivity of IVIg was significantly abolished when anti-HLA-E Abs were depleted, specifically by passing IVIg through HLA-E heavy chain-conjugated Affi-gel affinity columns. This suggested that IVIg’s HLA-Ia reactivity could possibly be due to the presence of HLA-E-specific IgG Abs, and that there were anti-HLA-E mAbs that might simulate the HLA reactivity of IVIg.

These observations led us to hypothesize that the anti-HLA-E mAbs that simulated the HLA-reactivity of IVIg could mimic IVIg by suppressing B-cells from producing Abs. We compared the efficacy of IVIg versus that of mAb TFL-007 (HLA-I polyreactive, anti-HLA-E mAb) (Table 4) in suppressing the B-cell blastogenesis, proliferation, and production of Abs.

#### 5.2.2. Methodology to Test the Hypothesis

More details are presented elsewhere [92]. Two commercial IVIg preparations were used. They were (1) IVIg-GamaSTAN™ (Lot 26NJ651; Talecris Biotherapeutics, Inc., Research Triangle Park, NC, USA) formulated as a 15–18% protein solution at a pH of 6.4–7.2 in 0.21–0.32 M glycine and, (2) IVIgGamunex^®^-C (Lot 26NKLG1, Talecris), a solution at a pH of 6.4–7.2 in 0.16–0.24 M glycine, albumin < 20 μg/mL. The HLA-I polyreactive HLA-E mAbs used were TFL-007s (culture supernatant) and TFL-007a (ascites). For suppressing antibody production by B-cells, two different experiments were carried out: 

Experiment # 1 was on freshly purified B-cells from the peripheral blood lymphocytes of a woman alloimmunized with her husband’s HLA DRB1*0101 antigens during her first pregnancy.

The HLA typing of the woman, her husband, and their two daughters (first, 23 years; second, 18 years) revealed that both the father and the first daughter carried the DRB1*0101 allele, while the mother’s sera showed the prevalence of high levels of allo-antibody against DRB1*0101. Periodic screening indicated that the high MFIs of the primary allo-HLA Abs had persisted for the past two years. Most likely, the anti-DRB1*0101 IgG Abs with high MFI were formed as a consequence of alloimmunization during a pregnancy that occurred 23 years before, which suggested the prevalence of both long-lived memory B-cells and bone marrow-resident plasma cells. Because this alloantibody was bound to the husband’s primary allele, the antibody was designated as the ‘primary alloantibody’. 

Therefore, her peripheral blood B-cells were isolated and purified after obtaining her informed consent and institutional (TFL) approval. The B-cells were purified from the PBMC by positive selection, using CD19 Pan B Dynabeads^®^ magnetic beads (Invitrogen, Life Technologies Corporation, Carlsbad, CA, USA), and detached using DETACHaBEAD^®^ CD19 (Invitrogen). Purified B-cells (>95% CD19+) were plated at 0.2 × 106/200 μL/well in a sterile, 96-well plate (ThermoFisher Scientific, Inc., Waltham, MA, USA) and cultured in Iscove’s Modified Dulbecco’s medium, containing HEPES, L-glutamine, and sodium pyruvate (Gibco-Invitrogen) supplemented with AB human serum (10%), recombinant human (rh) insulin (5 μg/mL), rh-transferrin (50 μg/mL), gentamycin (25 μg/mL), and 2-mercaptoethanol (50 μM). The resting B-cells were activated with rh-IL-2 (50 ng/mL), rhIL-4 (100 ng/mL), rhIL-6 (100 ng/mL), rhIL-10 (50 ng/mL), and human CD40 antibody (1 μg/mL) [92]. The B-cell population (CD19+) isolated from PBMC, using positive selection on day 0, consisted of a major fraction including naive B-cells (CD20+/CD27−/CD38+/−) (74.47%), B-memory cells (CD20+/CD27+/CD38−) (8.47%), and plasma cells (CD20−/CD27++/CD38++) (0.26%). These cells, upon activation by the selected battery of cytokines IL-2/IL-4/IL-6/IL-10/IL-21 (at a 1:4:4:2:2 ratio) and human CD40 antibody (1 μg/mL) for seven days, resulted in an increase in plasma cells from 0.26% on day 0 to 36.25% on day 7. On day 7, culture supernatants [10 μL] from each well were analyzed for anti-HLA, class II IgG alloAbs. Cells from the positive wells were further harvested, washed (3x), seeded, and activated again as above. On days 8 and 9, the culture supernatants were screened for Abs. After ascertaining the consistency of the MFIs of the Abs on days 8 and 9, the cells were pooled, washed (3X), and aliquoted. They were maintained in culture without any cytokine activators or anti-CD40 antibody for an additional 3 days. The culture supernatants [10 μL] from each well were analyzed for Abs at 0, 12, 24, 48, and 72 h.

On day 9, the cells were pooled, aliquoted, and maintained without the cytokine combo or anti-CD40 mAb. These wells were exposed to medium or IVIg (1/100, 1.5 mg/mL) or mAb TFL-007s (1/100, 5 μg/mL) for 72 h. To study the effect of IVIg on the secretion of allo-HLA IgG Abs, we used IVIg at a protein concentration 300-fold higher than that of the purified culture supernatant of TFL-007s (5 μg/mL) used in the treatment of B-cells in culture. The supernatants recovered from the respective wells were screened for the HLA-alloAbs.

Experiment # 2 was on the human hybridoma cell line HML16, generated from the resting B-cells of a multiparous woman.

The human hybridoma cell line, HML16, produced anti-HLA, class I alloAbs with differing MFIs: high against B*0702, B*8101, B*6701, and B*4201; and low against B*2708, B*2705, B*5501, B*5601, and B*8201. The cell line was cultured in RPMI-1640, as described earlier. The cells were seeded at 1000/100 μL/well in a Falcon 96-well, flat-bottomed plate and divided into different treatment groups: Group 1, medium control; Group 2, mouse IgG control; Group 3, mAb TFL-007 (ascites); Group 4, IVIg (GamaSTAN); and Group 5, IVIg (Gamunex-C). The following sub-groups were established: mouse IgG control (100 and 50 μg/mL); GamaSTAN–IVIg subgroups were at dilutions 1:10 (15 mg/mL), 1:20 (7.5 mg/mL), and 1:40 (3.75 mg/mL); Gamunex-IVIg subgroups were at dilutions 1:10 (10 mg/mL), 1:20 (5 mg/mL), and 1:40 (2.5 mg/mL); and mAb TFL-007a subgroups were at dilutions 1:10 (100 μg/mL), 1:20 (50 μg/mL), 1:40 (25 mg/mL), and 1:80 (12.5 μg/mL). Twenty μL of culture supernatant from each well were analyzed for allo-HLA Abs at 0 and 72 h.

The expression of CD19 was monitored using the fluorescein isothiocyanate (FITC)-labeled, anti-human CD19 (mAb HIB19). On days 0 and 7, both resting and activated B-cells were stained with phycoerythrin (PE) anti-human CD20 (mAb 2H7), peridinin chlorophyll (PerCP) antihuman CD27 (mAb 0323), and FITC anti-human CD38 (mAb HIT2) to examine the differential activation of B-cells. The source of the mAbs was from BioLegend. Prior to staining with Abs, Human TruStain FcX™ (BioLegend, San Diego, CA, USA) was used to block FcR-involved, unwanted staining.

#### 5.2.3. TFL-007 Suppressed HLA-Antibody Production Better Than That of IVIg

In the first experiment, both the sera of the woman and of those secreted by activated memory B-cells (culture supernatants) showed the presence of several alloAbs directed against DRB1*0102, DRB1*0404, DRB1*0405, DRB1*1402, and DRB1*0401, in addition to the primary alloantibody, anti-DRB1*0101 IgG. The secondary alloAbs were not directed against the husband’s alleles. They might have represented cross-reactive alloAbs in that they occurred in cultures containing the primary alloantibody, anti-DRB1*0101. These alleles shared amino acids or amino acid sequences with the primary allele, DRB1*0101. The details were presented elsewhere [92].

Figure 10 documents that the GamaSTAN IVIg suppressed the secretion of the primary alloantibody against DRB1*0101 at different time points (*p*^2^ < 0.01). Similarly, the HLA polyreactive mAb TFL-007s also significantly reduced the secretion of the primary alloantibody (*p*^2^ < 0.0005). However, the suppression of the secretion of the primary alloantibody by the mAb TFL-007s was highly significant compared to the IVIg-induced suppression.

In the second experiment, the treatment effects of two preparations of IVIg (GamaSTAN and Gamunex) were tested on HM16 at different dilutions and protein concentrations and compared with the media control [92]. The IVIg concentration used for the hybridoma cell line HML16 was comparable to the high-dose IVIg used for desensitization and post-transplant therapy. Neither of the IVIg preparations showed any significant suppression of the secretion of allo-HLA-B IgG by the hybridoma cells. In contrast, the mAb TFL-007a had a strikingly significant suppressive effect on the secretion of both HLA-B*0702 and B*8101. More importantly, the mAb TFL-007a showed dosimetric suppression of allo-HLA Abs. When HML16 was treated with the highest concentration of TFL-007a (100 μg/mL), suppression was 33% for B*0702 and 34% for B*8101, compared with the medium control group. When the dosage of TFL007a was decreased, the suppression effect declined. In short, in marked contrast to the IVIg preparations, the HLA-I polyreactive mAb TFL-007a significantly suppressed the secretion of both allo-HLA-B Abs.

### 5.3. Expansion of Foxp^3^+ Tregs In Vitro: IVIg versus HLA-I Polyreactive mAbs

Human CD4+CD25+Foxp^3+^ regulatory T-cells are a naturally occurring population of regulatory T-cells (Tregs) in circulation [93,94]. Their presence in liver allografts is attributed to tolerance of the transplanted organ [95,96,97,98,99,100,101]. They suppress Ab production by downregulating B memory and plasma cells [102] and depleting both CD4+ [103] and CD8+ [104,105] T-cells, and hence, they play a major role in graft tolerance [101,106,107]. While performing skin graft experiments on human-CD52 transgenic CP1-CBA/Ca (H-2^k^) mice, Garca et al. [106] observed that the “T cell suppression of graft rejection is an active process that operates beyond secondary lymphoid tissue, and involves the persistent presence of regulatory T-cells at the site of the tolerated transplant.” (p. 1641).

The therapeutic mAb, tocilizumab, given to patients with rheumatoid arthritis, increases T-regs and correlates with clinical response [108]. Upregulation of T-regs by IVIg is considered a critical factor in controlling experimental autoimmune encephalomyelitis; IVIg is known to upregulate T-regs [109].

Because IVIg can upregulate T-regs, it is hypothesized that the HLA-I polyreactive mAbs TFL-006 and TFL-007 may also induce proliferation of T-regs. For testing this hypothesis, CD4+CD25+foxp^3^+ Tregs were obtained from the peripheral blood of normal and healthy donors after obtaining necessary consent and institutional IRB approval. A variety of cell surface markers, including CD4, CD25 (IL-2Rα), CD45RA, and FoxP3, were monitored, using their respective monoclonal Abs.

We compared the impact of different commercial preparations of IVIg and the HLA-I polyreactive mAb TFL-007 in triplicate on untreated T-regulatory cells (CD4+/CD25+/FoxP^3+^) obtained from the blood of a healthy volunteer from TFL. The mAb purified from ascites was used throughout. Figure 11 illustrates that the different commercial preparations of IVIg at two different dilutions (1/10 and 1/80) failed to upregulate the Tregs, while the mAb TFL-007a showed a significant increase in the number of cells as compared to controls.

## 6. Discussion

IVIg is a mixture of polyclonal IgG Abs, pooled and purified from thousands of normal individuals. The binding affinity of IVIg is multivarious because the IgG Abs in IVIg can bind to blood groups, MHC complex antigens, cytokines, chemokines and their receptors, and several other antigens including human albumin. Most importantly, in the last three decades [2,3,49,50,51,52,53,54,55,56,57,82,83,84,85,86,87,88,89,90,91], IVIg has been administered to HLA-sensitized patients who possess a higher level of HLA allo-Abs, with the intention to lower these Abs. Due to the high level of these allo-HLA Abs, such patients are more likely to have a positive crossmatch result with potential organ donors pre-transplantation, and they often languish for years on the waiting list. For highly sensitized patients, the prospects of transplantation get grim. Realizing the imminent need to lower the impact of HLA sensitization, the US FDA approved IVIg for therapeutic administration in HLA-sensitized patients. However, due to its lack of efficacy as a monotherapy [55,56,57], IVIg is often combined with other therapeutic agents [89,90,91] such as plasmapheresis and rituximab (a mAb that depletes CD20+ B-cells). Commercial therapeutic preparations of IVIg, formulated in different countries, possess allo-HLA IgG Abs against almost all alleles of all six loci of HLA-I [35,57]. In this regard, HLA polyreactive mAbs developed from HLA-E αHCs mimic the allo-HLA antibody profile of IVIg [the U.S. Patent 10,800,847; 13 October 2020].

The structure of a single HLA-I antigen has two different profiles, as illustrated in Figure 2. The most well-known profile is the cell surface heteromer, consisting of α-HC associated with β2m (Figure 1). In the heteromeric profile, many of the most commonly shared amino acid sequences (Table 1 and Table 2), such as ^117^AYDGKDY^123^ and ^127^LNEDLRSWTA^135^ are cryptic, hidden by β2m. The second less-known profile of a single HLA-I antigen is a β2m-free a-HC, i.e., an OC (Figure 2). Although this structure is naturally occurring, it is often misconstrued to be identical to denatured HLA antigens, as found on alkali or acid-treated SABs [46,110]. In fact, the misconception that the monomeric versions of naturally occurring HLA molecules are “denatured” HLA is well-documented in several reports [13,14,15,16,17,18,19,20,21,22,23,24,25,26,27,28,29,30]. These reports demonstrate several novel aspects of the monomeric versions. The shared amino acid sequences common to all six loci of HLA are exposed for immune recognition and serve as receptors for signal transduction. Notably, the long cytoplasmic tails of the monomeric versions on cells with tyrosyl and seryl residues (Figure 2) are shown to be involved in signal transduction (Figure 9). Arosa et al. [19,111] elucidated further the difference between the intact HLA antigens, as CCs, and the monomeric variants, as OCs, when expressed in vivo on the cell surface (Figure 4). One difference was that HLA molecules shed from the cell surface [112] occurred as monomeric variants, or OCs, in the circulation (Figure 3). Since they exposed cryptic epitopes, they served as immunogens for allo-HLA Ab production. Indeed, the immunogenicity and antigenicity of the shared cryptic epitope ^117^AYDGKDY^123^ were much greater than other epitopes (Table 2).

Visualizing the antibody recognition sites (epitopes) of CCs and OCs would clarify and elucidate the HLA profile instrumental for generation of HLA-I polyreactive IgG mAbs and naturally occurring IgG Abs. Theoretically, one can expect the presence of polyreactive IgG Abs recognizing the OCs of HLA in IVIg. Indeed Figure 5 confirms the prevalence of two commercial preparations of IVIg reacting to monomeric variants on the acid denatured SABs. The density of IgG binding to monomeric variants was much higher than those recognizing the CCs on ‘iBeads’ (Figure 5). Kaveri et al. [59] pointed out “Abs to a conserved region (cryptic domain) of HLA class I molecules, capable of modulating CD8 T cell-mediated function, are present in pooled normal immunoglobulin for therapeutic use” (p. 865). Strikingly, HLA-polyreactive IgG2a mAbs (TFL-006 and TFL-007) do not recognize CCs, but recognize only the OCs on the SABs (Table 5). Importantly, as noted above, these naturally occurring OCs existing on the cell surface in vivo should not be considered synonymous with chemical treatment (acid or alkali) on synthetic beads disrupting intact CCs to produce denatured α-HCs in vitro.

The binding of the HLA-I polyreactive mAbs to OCs is responsible for the suppression of blastogenesis and proliferation. It may involve the reversal of phases of activation of T-lymphocytes, mediated by signal transduction. The elongation of the cytoplasmic tail of the HLA-I OCs exposes otherwise cryptic tyrosine-320 [76] and serine-335 [113] residues, with a provision for phosphorylation (Figure 9). Although serine-335 is generally considered the primary site of phosphorylation in this tail, the phosphorylation of tyrosine-320 has been indicated by others [76]. Thus, the HLA-I OCs in activated T-cells are associated with tyrosine phosphorylation and are capable of enabling cis interactions with cell surface receptors or other signaling molecules [76,77,113,114,115,116]. The binding by TFL mAbs may result in dephosphorylation of the cytoplasmic tails of CD3 molecules by activating phosphatases. The result is to arrest transcription factors and inhibit the synthesis of the proteins involved in blastogenesis and mitosis (Figure 9). These events suggest that the suppression of the activation of T-cells could be due to the binding of HLA-I polyreactive mAbs, mimicking the HLA-I polyreactivity of IVIG, to the shared amino acid sequences exposed on the naturally occurring OCs of HLA-I molecules (Figure 2). This activation involves not one HLA-I locus but all alleles of the six loci (HLA-A, HLA-B, HLA-C, HLA-E, HLA-F, and HLA-G) expressed on the surface of activated T-cells. The efficacy of the monoclonality of the HLA-I polyreactive mAbs and their F(ab′)2 binding is restricted to OCs. Furthermore, poor efficacy of IVIg is due to admixture with CC-binding Abs, other non-HLA Abs, anti-idiotypic Abs, immune complexes, and other chemicals such as sugars added to render stability to IVIg.

Possibly, the same or similar phosphorylation and dephosphorylation mechanisms may be involved in suppressing antibody production by B-cells. The HLA-I polyreactive mAb, but not IVIg, suppressed allo-HLA-I and allo-HLA-II Abs production by B-memory cells and immortalized B-cells. While monitoring the efficacy of the HLA-I polyreactive mAbs, the cytokine combination and anti-CD40 antibody were both removed to precisely evaluate the impact of the antibody. Possibly, HLA-I polyreactive mAbs may also help to suppress other Abs produced by B-cells and alloAbs as well.

In addition, the HLA-I polyreactive mAbs upregulated T-regs better than IVIg (Figure 11). T-regs are well known for their immunoregulatory properties. They suppress Ab production by downregulating B-memory and plasma cells [102] and by depleting CD4+ [103] and CD8+ [104,105] T-cells. We believe that both IVIg and the HLA-I polyreactive TFL mAbs perform the immunosuppressive functions stated above, including the upregulation of T-regs, by binding to the OCs of HLA.

## 7. Conclusions

The observations point out conclusively that IVIg (a) suppressed blastogenesis and proliferation of T-cells, (b) minimized the HLA-II allo-antibody production by B-memory cells of parous women and HLA-I antibody production of an immortalized cell line, and (c) expanded T-regs. These effects could be due to the HLA OC-polyreactive Abs present in IVIg. These findings are further strengthened by the several similarities observed in the nature and functional characteristics of IVIg and HLA-I polyreactive, OC-specific TFL mAbs (TFL-006 and TFL-007) (Table 7). The mAbs, which recognize shared epitopes on naturally occurring OCs, with no ability to bind to the CCs, perform the same suppressive functions better than the different preparations of IVIg. Possibly, anti-CC Abs present in IVIg may impact their above-mentioned immunomodulatory functions. Hence, HLA-I polyreactive TFL mAbs are capable of serving as IVIg-mimetics, perhaps more efficiently than IVIg itself. Clinical trials are clearly warranted. Possibly, humanized versions of these TFL-mAbs, either combined or alone, can be better therapeutic tools than IVIg to suppress HLA sensitization and minimize antibody production post-transplantation.

## Figures and Tables

**Figure 1 vaccines-09-00680-f001:**
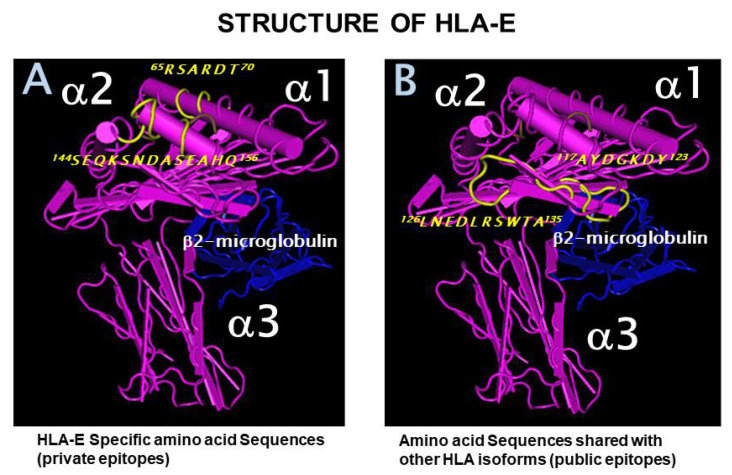
HLA-E structure shows specific amino acid sequences, private epitopes, shared amino acid sequences, or public epitopes. (**A**). Private epitopes recognized by the anti-HLA-E monospecific mAb, TFL-033 (**B**). Public epitopes are recognized by the anti-HLA-E mAbs, TFL-006, and TFL-007. Both private and public epitopes are shown in yellow.

**Figure 2 vaccines-09-00680-f002:**
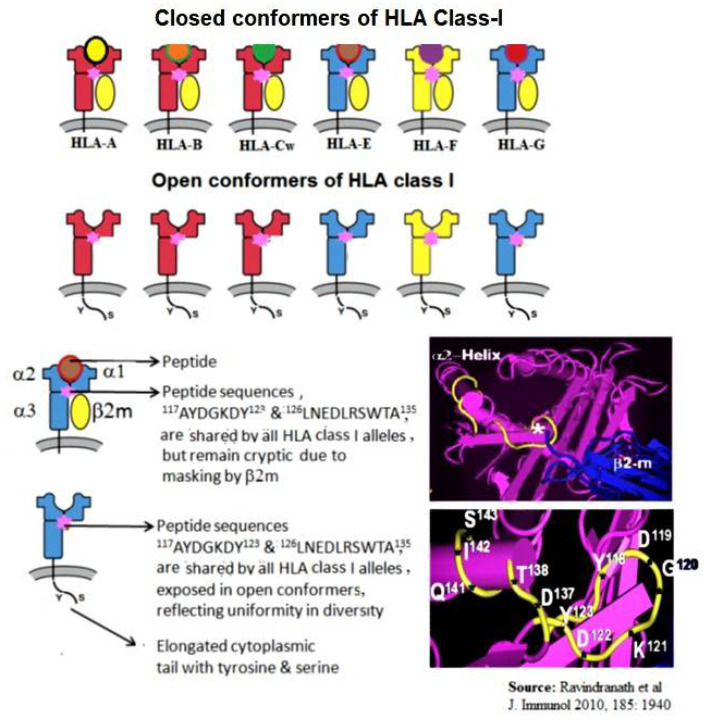
Diagrammatic illustration of six isoforms of HLA class I molecules. The upper row shows the structures of intact, trimeric, closed conformers (CCs) of HLA. Note the shortened cytoplasmic tail. The lower row shows monomeric HLA-I (open conforms OCs). OCs of HLA-I expose epitopes masked by β2m. These epitopes, particularly ^117^AYDGKDY^123^ and ^126^LNEDLRSWTA^135^, are found in almost all alleles of the six isoforms of HLA.

**Figure 3 vaccines-09-00680-f003:**
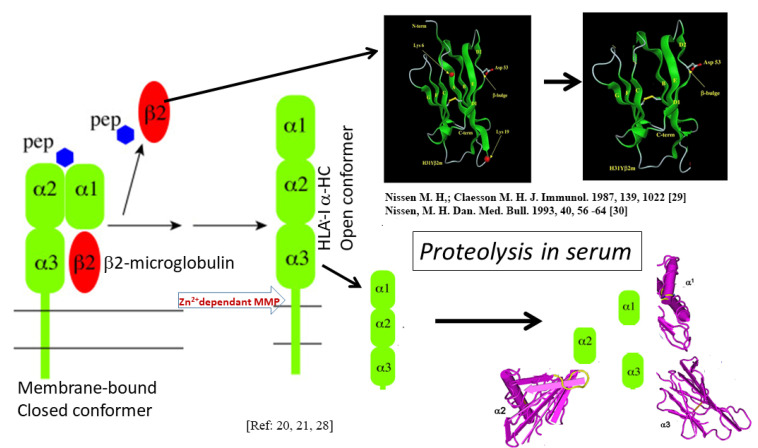
Shedding of membrane-bound CC and the shedding results in dissociation of HLA α-HC from β2m. The membrane-bound α-HC is cleaved by a Zn^2+^-dependent membrane metalloproteinase. The shed α-HC and β2m are further degraded. Immune recognition of shed β2m and α-HC can occur at any time [29,30].

**Figure 4 vaccines-09-00680-f004:**
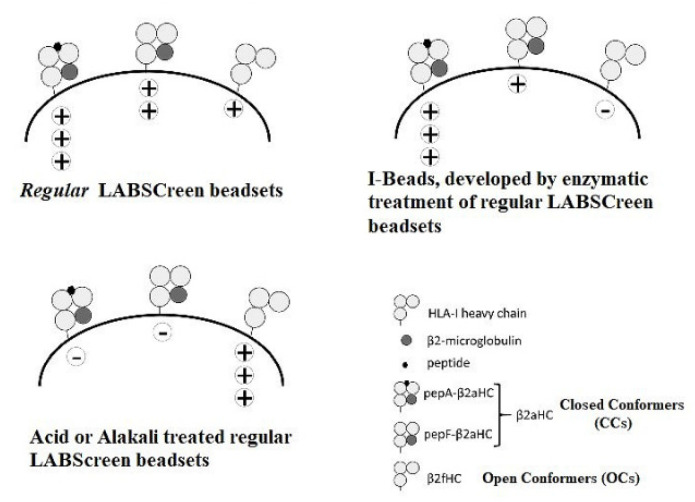
Three different kinds of microbeads used: Regular beads admixed with CCs and OCs; The acid or alkali treated regular beads with OC only, (iii) beads restricted with CC. The microbeads were characterized using W6/32 (IgG2a) bound to peptide-associated and peptide-free CCs, but not bound to the OCs. HC-10 (IgG2a) recognized CCs devoid of a peptide. HLA-I polyreactive mAb TFL-006 (IgG2a) bound to OCs only and was inhibited by the shared peptides (^117^AYDGKDY^123^ and ^126^LNEDLRSWTA^135^) found in all HLA-I isoforms.

**Figure 5 vaccines-09-00680-f005:**
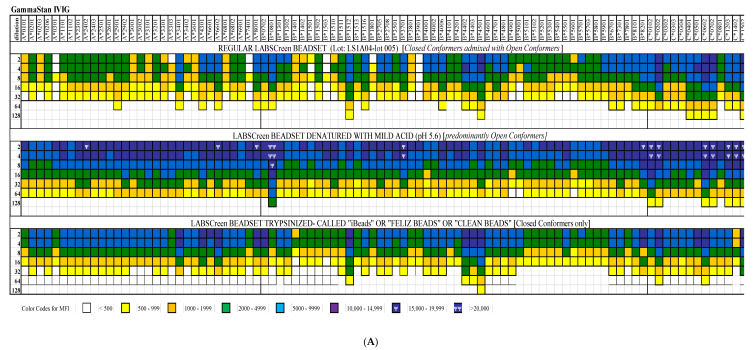
HLA-I polyreactivity of IVIg. IVIg was tested on three kinds of SABs: (i) regular LABScreen bead set coated with an admixture of CCs and OCs, (ii) regular LABScreen bead set treated with a mild acid to convert OCs to CCs. The bead set was loaded with OCs. (iii) “iBeads” generated from regular LABScreen bead sets to enzymatically eliminate OCs, and therefore containing only CCs. The Luminex immunoassay with IVIg from GammaSTAN (**A**) and Octagam (**B**) confirmed the HLA-I polyreactivity of IVIg. Importantly, IVIG reacted with the monomeric variants on the acid-denatured SABs better than on iBeads. The density of IgG binding is illustrated by mean fluorescent intensity (MFI), as shown in different colors. The binding affinity of IVIg to monomeric variants was much higher than those recognizing the CC on ‘iBeads.’.

**Figure 6 vaccines-09-00680-f006:**
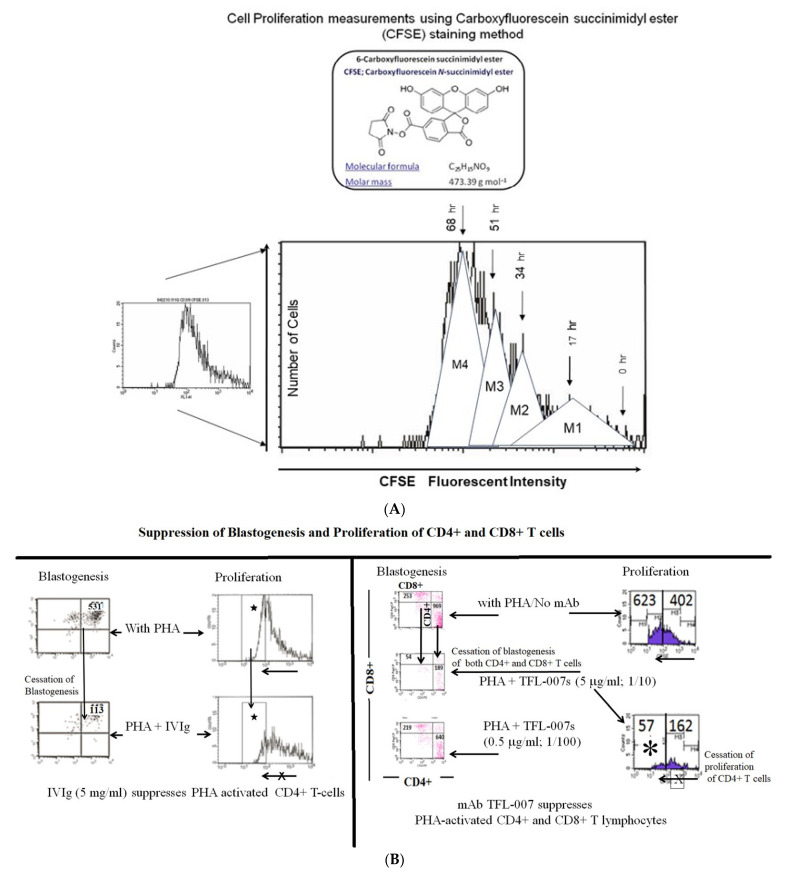
(**A**). The proliferation assay is based on labeling the purified T-cells during PHA activation with the intracellular fluorescent dye carboxyfluorescein succinimidyl ester (CFSE: C25H15NO9; mol. mass: 473.39 g/mol) and using flow cytometry, measuring mitotic activity by the successive twofold reductions in fluorescent intensity of the T-cells placed in culture for 72 h. CFSE is cell-permeable and is retained for long periods within cells by covalently coupling by means of its succinimidyl group to intracellular molecules. Due to this stable linkage, once incorporated within cells, CFSE is not transferred to adjacent T-cells but remains in the cell even after several mitotic divisions. (**B**). Suppression of blastogenesis and proliferation of CD4+ T-cells by IVIg (Globex) and HLA-I polyreactive mAb TFL-007 at similar protein concentrations. The CFSC profile illustrates suppression as indicated by asterisks.

**Figure 7 vaccines-09-00680-f007:**
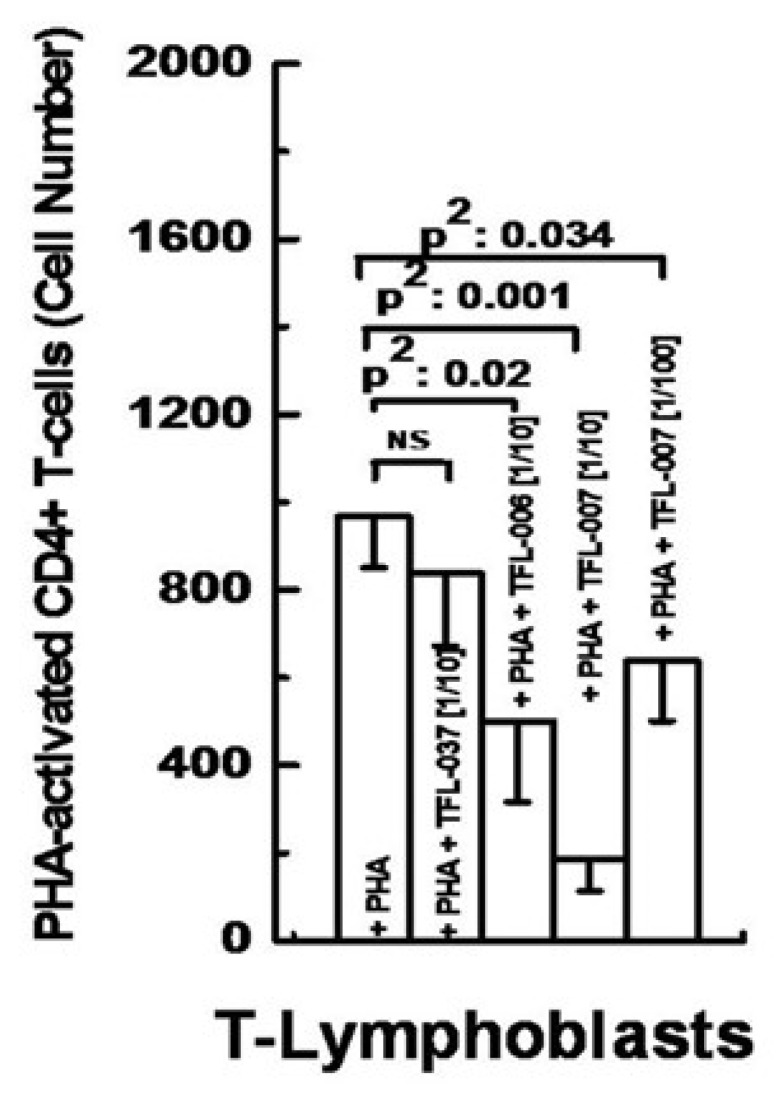
Suppression of PHA-activated CD4+ T-cells by HLA polyreactive monoclonals TFL-006 and TFL-007 at 1/10 dilutions. The control mAb, TFL-037, failed to suppress PHA-activated proliferation, whereas the HLA-I polyreactive mAbs suppressed proliferation significantly. Dosimetric suppression of TFL-007 is shown.

**Figure 8 vaccines-09-00680-f008:**
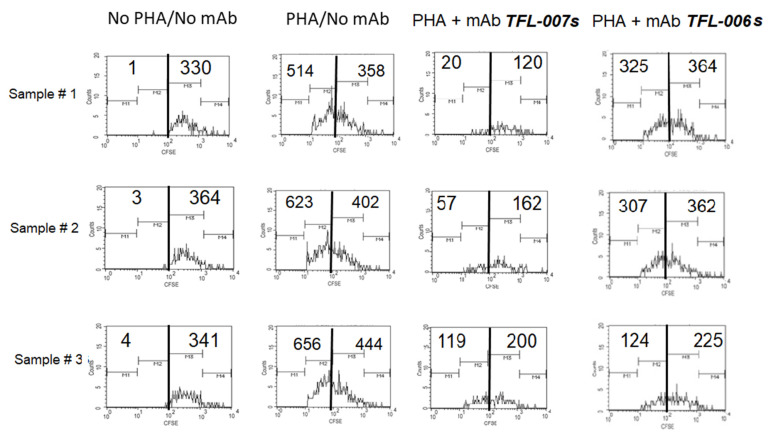
Suppression of proliferation of PHA-activated CD4+ T-cells by purified culture supernatant of anti-HLA-E mAb TFL-006s. Results of the triplicate analysis are presented. The numbers inside refer to cell counts. Refer to Figure 5A to understand the meaning of the shift in the profiles from right to left.

**Figure 9 vaccines-09-00680-f009:**
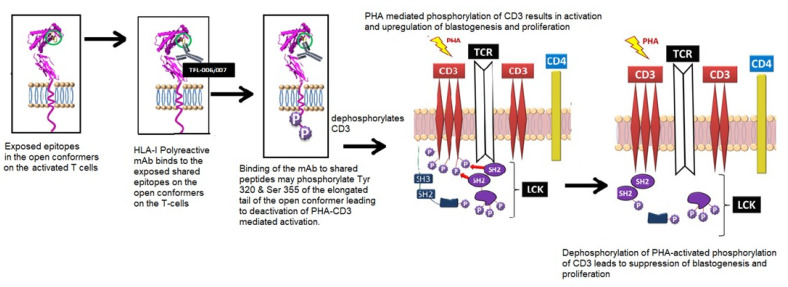
A model illustrating the possible mechanism underlying phytohemagglutinin (PHA) activation of T-cells and the suppression of activated T-cells mediated by HLA-I polyreactive mAbs (TFL-006 and TLF-007) and possibly by IVIg. The model is based on a model proposed by Mustelin, Vang, and Bottini [77] for T-cell activation. The CD3/T-cell receptor (TCR)/CD4 structure on the lipid raft (pink zone) of the bi-layered lipid membrane on the non-phosphorylated, non-activated CD4+ T-cells is illustrated. The lymphocyte-specific protein, tyrosine kinase (LCK), induced phosphorylation of tyrosine-based activation in the cytoplasmic domain of CD3, which led to the activation of transcription factors and the transcription of cell surface molecules such as interleukin (IL)-2Rα and open conformers of HLA class I. SH-1, SH-2, and SH-3 represent family members of Src homology; they are involved in mediating the cytoplasmic domain of CD3. Further activation of the tyrosyl-phosphorylated motifs and then interaction with SH-1 domains within the protein kinase LCK led to further signaling function [77]. Importantly, the exposure of shared amino acid sequences of all the HLA open conformers is indicated by a blue circle. It is this site that is recognized by TFL-006 and TFL-007. Possible interactions and consequences of recognition of the shared peptide sequences by the HLA-I polyreactive IgG mAbs are illustrated in three steps: first, the exposure of the shared peptide sequence on the open conformer; secondly, recognition of the shared epitopes on the open conformer by the mAbs; thirdly, possible phosphorylation of the elongated cytoplasmic tail of open conformers. That elongation resulted in the exposure of cryptic tyrosine (Tyr320) and serine (Ser355) residues in the cytoplasmic tail. It might have been the binding of the mAbs to the shared peptide sequences that initiated the phosphorylation, leading to signal transduction. A final step involved initiation of dephosphorylation of the cytoplasmic domain of CD3, resulting in arrest of activation or suppression. That seemed plausible, as the phosphorylation was known to be reversible.

**Figure 10 vaccines-09-00680-f010:**
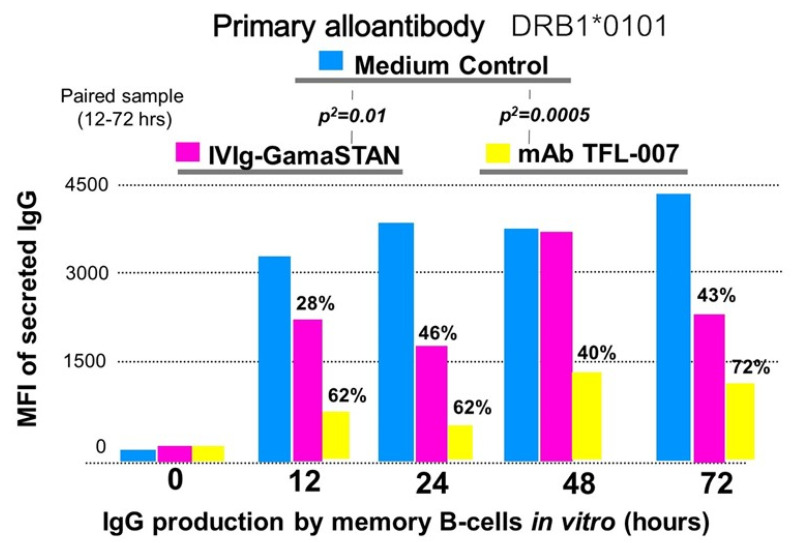
The impact of IVIg (GamaSTAN) and HLA-I polyreactive mAb (TFL-007) on the primary alloantibody DRB1*0101 (IgG), secreted by activated B-lymphocytes obtained from an alloimmunized woman. GamaSTAN S/D IVIg was used at 1:100 dilution, 1.5 mg. protein/mL. At the time when the IVIg was added, the cytokine combo and anti-CD40 antibody were not added. In all panels, the mean fluorescent intensity (MFI) of alloantibody secretion is compared between the medium control and treatment with IVIg and HLA-I polyreactive IgG2a mAb TFL-007. IVIg inhibited the secretion of the primary alloantibody at a significant level (*p*^2^ = 0.01). HLA-I polyreactive mAb inhibited the secretion of the primary alloantibody at a higher significant level (*p*^2^ = 0.0005). This is an original figure.

**Figure 11 vaccines-09-00680-f011:**
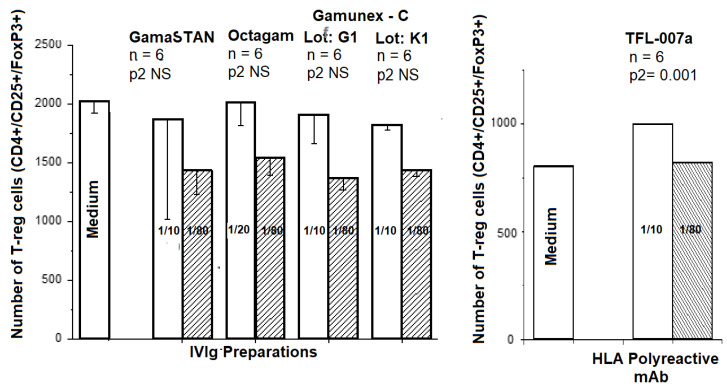
Effects of different commercial preparations of IVIg, which included GamaSTAN™ S/D (15–18 gm%, Lot 26NHCVI; (Talecris Biotherapeutics, Inc., Research Triangle Park, NC, USA) at dilutions 1/10 (concentration 15 mg/mL) and 1/80 (concentration 1.2 mg/mL), octagam^®^ (6 gm%, Lot A913A8431; Octapharma Pharmazeutika) at dilutions 1/20 (concentration 3 mg/mL) and 1/80 (concentration 0.75 mg/mL), Gamunex^®^-C (10 gm%, Lots 26NKLG1 and 26NKLK1; Telacris) at dilutions 1/10 (concentration 10 mg/mL) and 1/80 (concentration 8 mg/mL), and mAb TFL-007a (at dilution 1/10, concentration 62.7 μg/mL; 1/80, concentration 7.84 μg/mL) on PHA-untreated cells were compared with the effect of medium alone on the proliferation of Treg cells, defined as CD4+/CD25+/Foxp^3^+. Note that IVIg preparations used in this study failed to upregulate Tregs, in contrast to TFL-007a, which significantly upregulated Treg cells at dilution 1/10. This figure is original but derived and modified from a previous report [48].

**Table 1 vaccines-09-00680-t001:** Amino acid sequences of HLA-E compared with HLA-F and HLA-G. There are several sequences restricted to HLA-E, as illustrated by a short sequence, ^66^SARDT^70^, and a long sequence, ^143^SEGKSNDASE^152^. HLA-E shares several sequences with HLA-F and HLA-G. In addition, HLA-E may selectively share several sequences with HLA-G and a few sequences (^175^GKETL^179^ and ^193^PISDH^197^) with HLA-F. Above all, HLA-E shares ^117^AYDGKDY^123^ and ^126^LNEDLRSWTA^135^ with all other HLA isoforms (HLA-A, HLA-B, HLA-C, HLA-F, and HLA-G).

	1	2	3	4	5	6	7	8	9	10	11	12	13	14	15	16	17	18	19	20	21	22	23	24	25	26	27	28	29	30	31	32	33	34	35	36	37	38	39	40	41	42	43	44	45	46	47	48	49	50
**HLA-E**	**G**	**S**	**H**	**S**	L	K	Y	F	H	T	S	**V**	**S**	**R**	**P**	**G**	**R**	**G**	**E**	**P**	**R**	F	I	S	V	G	**Y**	**V**	**D**	**D**	**T**	**Q**	**F**	**V**	**R**	**F**	**D**	N	D	A	A	S	P	R	M	V	**P**	**R**	**A**	**P**
**HLA-G**	**G**	**S**	**H**	**S**	M	R	Y	F	S	A	A	**V**	**S**	**R**	**P**	**G**	**R**	**G**	**E**	**P**	**R**	F	I	A	M	G	**Y**	**V**	**D**	**D**	**T**	**Q**	**F**	**V**	**R**	**F**	**D**	S	D	S	A	C	P	R	M	E	**P**	**R**	**A**	**P**
**HLA-F**	**G**	**S**	**H**	**S**	L	R	Y	F	S	T	A	**V**	**S**	**R**	**P**	**G**	**R**	**G**	**E**	**P**	**R**	Y	I	A	V	E	**Y**	**V**	**D**	**D**	**T**	**Q**	**F**	L	R	F	D	S	D	A	A	I	P	R	M	E	P	R	E	P
						a1	a1	a1	a1	a1	a1	a1	a1	a1	a1	a1	a1	a1	a1	a1	a1	a1	a1	a1	a1	a1	a1	a1	a1	a1	a1	a1	a1	a1	a1															
	51	52	53	54	55	56	57	58	59	60	61	62	63	64	65	66	67	68	69	70	71	72	73	74	75	76	77	78	79	80	81	82	83	84	85	86	87	88	89	90	91	92	93	94	95	96	97	98	99	100
**HLA-E**	**W**	M	**E**	**Q**	**E**	**G**	S	E	Y	W	D	R	E	T	R	***S***	***A***	***R***	***D***	***T***	A	Q	L	F	R	V	N	L	R	**T**	**L**	**R**	**G**	**Y**	**Y**	**N**	**Q**	**S**	**E**	**A**	G	**S**	**H**	**T**	**L**	**Q**	W	M	H	G
**HLA-G**	**W**	V	**E**	**Q**	**E**	**G**	P	E	Y	W	E	E	E	T	R	N	T	K	A	H	A	Q	T	D	R	M	N	L	Q	**T**	**L**	**R**	**G**	**Y**	**Y**	**N**	**Q**	**S**	**E**	**A**	S	**S**	**H**	**T**	**L**	**Q**	W	M	I	G
**HLA-F**	W	V	**E**	**Q**	**E**	**G**	P	Q	Y	W	E	W	T	T	G	Y	A	K	A	N	A	Q	T	D	R	V	A	L	R	N	L	L	R	R	**Y**	**N**	**Q**	**S**	**E**	**A**	G	**S**	**H**	**T**	**L**	**Q**	G	M	N	G
																	a2	a2	a2	a2	a2	a2	a2	a2	a2	a2	a2	a2	a2	a2	a2	a2	a2	a2	a2	a2	a2	a2	a2	a2	a2	a2	a2	a2	a2	a2	a2	a2	a2	a2
	101	102	103	104	105	106	107	108	109	110	111	112	113	114	115	116	117	118	119	120	121	122	123	124	125	126	127	128	129	130	131	132	133	134	135	136	137	138	139	140	141	142	143	144	145	146	147	148	149	150
**HLA-E**	C	E	L	G	P	D	R	R	F	**L**	**R**	**G**	**Y**	E	Q	F	**A**	**Y**	**D**	**G**	**K**	**D**	**Y**	L	T	**L**	**N**	**E**	**D**	**L**	**R**	**S**	**W**	**T**	**A**	V	**D**	**T**	**A**	**A**	**Q**	**I**	***S***	***E***	***G***	***K***	***S***	***N***	***D***	***A***
**HLA-G**	C	D	L	G	S	D	G	R	L	**L**	**R**	**G**	**Y**	E	Q	Y	**A**	**Y**	**D**	**G**	**K**	**D**	**Y**	L	A	**L**	**N**	**E**	**D**	**L**	**R**	**S**	**W**	**T**	**A**	A	**D**	**T**	**A**	**A**	**Q**	**I**	S	K	R	K	C	E	A	A
**HLA-F**	C	D	M	G	P	D	G	R	L	**L**	**R**	**G**	**Y**	H	Q	H	**A**	**Y**	**D**	**G**	**K**	**D**	**Y**	I	S	**L**	**N**	**E**	**D**	**L**	**R**	**S**	**W**	**T**	**A**	A	D	T	V	A	Q	I	T	Q	R	F	Y	E	A	E
																	**SHARED WITH HLA-A/-B & -C**									**SHARED WITH HLA-A, HLA-B & HLA-C**																								
	a2	a2	a2	a2	a2	a2	a2	a2	a2	a2	a2	a2	a2	a2	a2	a2	a2	a2	a2	a2																														
	151	152	153	154	155	156	157	158	159	160	161	162	163	164	165	166	167	168	169	170	171	172	173	174	175	176	177	178	179	180	181	182	183	184	185	186	187	188	189	190	191	192	193	194	195	196	197	198	199	200
**HLA-E**	***S***	***E***	A	E	H	Q	**R**	**A**	**Y**	**L**	**E**	D	**T**	**C**	**V**	**E**	**W**	**L**	**H**	K	Y	L	E	K	**G**	**K**	**E**	**T**	**L**	L	H	L	E	P	**P**	**K**	**T**	**H**	**V**	**T**	**H**	**H**	**P**	**I**	**S**	**D**	**H**	**E**	**A**	**T**
**HLA-G**	N	V	A	E	Q	R	**R**	**A**	**Y**	**L**	**E**	G	**T**	**C**	**V**	**E**	**W**	**L**	**H**	R	Y	L	E	N	G	K	E	M	L	**Q**	**R**	**A**	**D**	**P**	**P**	**K**	**T**	**H**	**V**	**T**	**H**	**H**	P	V	F	D	Y	**E**	**A**	**T**
**HLA-F**	E	Y	A	E	E	F	R	T	Y	L	E	G	E	C	L	E	L	L	R	R	Y	L	E	N	**G**	**K**	**E**	**T**	**L**	**Q**	**R**	**A**	**D**	**P**	P	K	A	H	V	A	H	H	**P**	**I**	**S**	**D**	**H**	**E**	**A**	**T**
	201	202	203	204	205	206	207	208	209	210	211	212	213	214	215	216	217	218	219	220	221	222	223	224	225	226	227	228	229	230	231	232	233	234	235	236	237	238	239	240	241	242	243	244	245	246	247	248	249	250
**HLA-E**	**L**	**R**	**C**	**W**	**A**	**L**	**G**	**F**	**Y**	**P**	**A**	**E**	**I**	T	**L**	**T**	**W**	**Q**	Q	D	G	E	G	H	T	Q	D	T	**E**	**L**	**V**	**E**	**T**	**R**	**P**	**A**	**G**	**D**	**G**	**T**	**F**	**Q**	**K**	**W**	**A**	**A**	**V**	**V**	**V**	**P**
**HLA-G**	**L**	**R**	**C**	**W**	**A**	**L**	**G**	**F**	**Y**	**P**	**A**	**E**	**I**	I	**L**	**T**	**W**	**Q**	R	D	G	E	D	Q	T	Q	D	V	**E**	**L**	**V**	**E**	**T**	**R**	**P**	**A**	**G**	**D**	**G**	**T**	**F**	**Q**	**K**	**W**	**A**	**A**	**V**	**V**	**V**	**P**
**HLA-F**	**L**	**R**	**C**	**W**	**A**	**L**	**G**	**F**	**Y**	**P**	**A**	**E**	**I**	T	**L**	**T**	**W**	**Q**	R	D	G	E	E	Q	T	Q	D	T	**E**	**L**	**V**	**E**	**T**	**R**	**P**	**A**	**G**	**D**	**G**	**T**	**F**	**Q**	**K**	**W**	**A**	**A**	**V**	**V**	**V**	**P**

**Table 2 vaccines-09-00680-t002:** Comparison of the amino acid sequence of HLA-E with sequences of five other HLA isoforms. Several peptide sequences of HLA-E are shared with the alleles of other HLA-I isoforms. Note that amino acid sequence AYDGKDY is shared with the maximum number of alleles of all isoforms of HLA-I, while sequences PRAPWMEQE and EPPKTHVT are shared with one allele of HLA-A (A*3306) and one allele of HLA-B (B*8201). The bioinformatics analysis was carried out using the Immune Epitope Database (IEDB) to predict the antigenicity rank of epitopes. The Chou and Fasman beta turn, Kolaskar and Tongaonkar antigenicity, Karplus and Schulz flexibility, and Parker hydrophilicity prediction methods in IEDB were employed. The methods predict the probability of specific sequences in HLA-E that bind to Abs being in a beta turn region, being antigenic, being flexible, or being in a hydrophilic region. Antigenicity rank is calculated by pooling the probability values.

HLA-E Peptide Sequences	HLA Alleles						Method 1	Method 2	Method 3	Method 4	Rank of Antigenicity
							Prediction SCORES				
	Classical HLA-Ib			Non-Classical HLA-Ib		Specificity	Beta-Turn	Antigenicity	Flexibility	Hydrophilicity	
[total number of amino acids]	A	B	Cw	F	G		Chou & Fasman (1978)	Kolaskar & Tangaonkar (1990)	Karplus & Schulz (1985)	Parker (1986)	
^47^PRAPWMEQE^55^ [9]	1	0	0	0	0	A*3306	0.993	0.948	0.969	0.586/1.143/1.657	
^58^EYWDRETR^65^ [8]	5	0	0	0	0	A restricted	0.993	0.915	1.024	3.301/2.786	10
^90^AGSHTLQW^97^ [8]	1	10	48	0	0	Polyspecific	1.019	1.033	0.989	2.629/0.901	6
^108^RFLRGYE^114^ [7]	24	0	0	0	0	A restricted	0.933	0.996	0.996	0.229	8
^115^QFAYDGKDY^123^ [9]	1	104	75	0	0	Polyspecific	1.059	1.001	0.993	2.629/3.201	5
**^117^AYDGKDY^123^** [7]	**491**	**831**	**271**	**21**	**30**	**Polyspecific**	**1.204**	**0.989**	**1.061**	**4.243**	1
^126^LNEDLRSWTA^135^ [10]	239	219	261	21	30	Polyspecific	1.046	0.983	1.039	2.443/2.329	2
^137^DTAAQI^142^ [6]	0	824	248	0	30	Polyspecific	0.813	1.065	0.978	1.957	3
^137^DTAAQIS^143^ [7]	0	52	4	0	30	Polyspecific	0.946	1.012	0.97	3.414	7
^157^RAYLED^162^ [6]	0	1	0	0	0	B*8201	0.929	0.996	0.969	2.601	
^163^TCVEWL^168^ [6]	282	206	200	0	30	Polyspecific	0.841	1.115	0.929	−0.914	4
^183^EPPKTHVT^190^ [8]	0	0	19	0	0	C restricted	1.029	1.044	1.042	3.043	9
^65^RSARDTA^71^ [7]	0	0	0	0	0	E restricted	1.011	0.952	1.038	4.901	2
^143^SEQKSNDASE^152^ [10]	0	0	0	0	0	E restricted	1.231	0.923	1.222	7.071/6.443/6.257/6.514	1

Method 1. Predict beta turns in protein secondary structures. Chou PY, Fasman GD. Prediction of the secondary structure of proteins from their amino acid sequence. Adv Enzymol Relat Areas Mol Biol. 1978;47:45-148. DOI: 10.1002/9780470122921.ch2. Method 2. A semi-empirical method which made use of the physicochemical properties of amino acid residues and their frequencies of occurrence in experimentally known segmental epitopes was developed to predict antigenic determinants on proteins. Application of this method to a large number of proteins has shown that the method can predict antigenic determinants with about 75% accuracy, which is better than most of the known methods. Kolaskar AS, Tongaonkar PC. A semi-empirical method for prediction of antigenic determinants on protein antigens. FEBS Lett. 1990 Dec 10;276(1-2):172-4. doi: 10.1016/0014-5793(90)80535-q. Method 3. In this method, a flexibility scale, based on the mobility of protein segments on the basis of the known temperature B factors of the a-carbons of 31 proteins of known structure, was constructed. The calculation based on a flexibility scale is similar to classical calculation, except that the center is the first amino acid of the six amino acids’ window length, and there are three scales for describing flexibility instead of a single one. Karplus PA, Schulz GE. Prediction of Chain Flexibility in Proteins—A tool for the Selection of Peptide Antigens. Naturwissenschafren 1985; 72:212-3. Method 4. In this method, a hydrophilic scale based on peptide retention times during high-performance liquid chromatography (HPLC) on a reversed-phase column was constructed. A window of seven residues was used for analyzing epitope region. The corresponding value of the scale was introduced for each of the seven residues and the arithmetical mean of the seven residue values was assigned to the fourth, (i+3), residue in the segment. Parker JM, Guo D, Hodges RS. New hydrophilicity scale derived from high-performance liquid chromatography peptide retention data: correlation of predicted surface residues with antigenicity and X-ray-derived accessible sites. Biochemistry. 1986 Sep 23; 25(19):5425-32.

**Table 3 vaccines-09-00680-t003:** The HLA-1 signatures of the mAbs generated after immunizing recombinant heavy chains of HLA-E. Group 10 is truly befitting the definition of a polyreactive mAb category.

IMMUNOGEN HLA-E^R107^ OR HLA-E^G107^							
Groups	Number	HLA-CLASS Ib			HLA-CLASS IC		
of mAbs	of mAbs	HLA-E	HLA-F	HLA-G	HLA-A	HLA-B	HLA-C
Group 1	24	+	−	−	−	−	−
Group 2	1	+	+	−	−	−	−
Group 3	1	+	−	+	−	−	−
Group 4	8	+	+	+	−	−	−
Group 5	4	+	−	−	−	+	−
Group 6	31	+	−	−	−	+	+
Group 7	109	+	−	−	+	+	+
Group 8	11	+	+	−	+	+	+
Group 9	18	+	−	+	+	+	+
Group 10	7	+	+	+	+	+	+

**Table 4 vaccines-09-00680-t004:** HLA-I allele reactivities of the polyreactive mAbs as compared with monospecific mAbs. The values represent mean fluorescent intensities (MFIs) of the mAbs, corrected against background values. The mAbs were generated using HLA-E-recombinant heavy chains. The number of HLA antigens showing positive reactivity with the mAbs are shown in bold letters.

*mAbs*	*Monospecific*	*Polyreactive*		*mAbs*	*Monospecific*	*Polyreactive*		*mAbs*	*Monospecific*	*Polyreactive*	
	TFL-033	TFL-006	TFL-007		TFL-033	TFL-006	TFL-007		TFL-033	TFL-006	TFL-007
	*IgG1*	*IgG2a*	*IgG2a*		*IgG1*	*IgG2a*	*IgG2a*		*IgG1*	*IgG2a*	*IgG2a*
**Neg**	3	15	7	***B* alleles***				***C* alleles***			
**Pos**	71	88	85	**B*0702**		1331	841	**C*0102**		7242	3268
**HLA-E**	**24411**	**22522**	**21618**	**B*0801**		2092	1033	**C*0202**		10690	6084
**HLA-F**		**12650**	**11035**	**B*1301**		5654	3979	**C*0302**		5917	3062
**HLA-G**		7193	2670	**B*1302**		2237	1426	**C*0303**		7114	4250
***A* alleles***				**B*1401**		11319	8767	**C*0304**		6584	3891
**A*0101**		2395	1037	**B*1402**		4414	2558	**C*0401**		2843	1272
**A*0201**		856		**B*1501**		1097		**C*0501**		16131	13096
**A*0203**		1095		**B*1502**		6256	4497	**C*0602**		9396	4274
**A*0206**		1494	843	**B*1503**		2831	1926	**C*0702**		12251	6919
**A*0301**		818		**B*1510**		2616	1470	**C*0801**		13456	10733
**A*1101**		10190	8476	**B*1511**		9041	5902	**C*1203**		5055	2102
**A*1102**		860		**B*1512**		1624	996	**C*1402**		8727	4936
**A*2301**		614		**B*1513**		5326	3365	**C*1502**		6030	3225
**A*2402**		3133	2011	**B*1516**		5614	3443	**C*1601**		8462	4364
**A*2403**		3151	1967	**B*1801**		6990	4890	**C*1701**		13521	9069
**A*2501**		1230	692	**B*2705**		2591	1576	**C*1802**		17918	15207
**A*2601**		3368	1638	**B*2708**		4437	2671	***C* alleles***	***0***	***16***	***16***
**A*2901**		3194	2256	**B*3501**		10205	8594				
**A*2902**		2235	1136	**B*3701**		6472	4338				
**A*3001**		2229	1237	**B*3801**		3844	1820				
**A*3002**		3353	2211	**B*3901**		7093	5304				
**A*3101**		858		**B*4001**		5743	3758				
**A*3201**		2237	1508	**B*4002**		6118	4675				
**A*3301**		2791	1627	**B*4006**		15643	13758				
**A*3303**		4212	2961	**B*4101**		7191	5277				
**A*3401**		6268	3968	**B*4201**		636					
**A*3402**		1399	893	**B*4402**		7062	4059				
**A*3601**		5806	3826	**B*4403**		7256	5638				
**A*4301**		4420	2364	**B*4501**		9535	7646				
**A*6601**		3644	1526	**B*4601**		6491	4130				
**A*6602**		1395	789	**B*4701**		6528	3895				
**A*6801**		1314	859	**B*4801**		4365	2716				
**A*6802**		2078	1276	**B*4901**							
**A*6901**		1964	917	**B*5001**		741					
**A*7401**		723		**B*5101**		6205	3724				
**A*8001**		2841	1430	**B*5102**		5251	3579				
***A* alleles***	***0***	***32***	***24***	**B*5201**		4524	2728				
				**B*5301**		8807	7323				
				**B*5401**		5556	4153				
				**B*5501**		2829	1887				
				**B*5601**		1386	777				
				**B*5701**							
				**B*5703**		1229	600				
				**B*5801**		10160	8047				
				**B*5901**		5646	3001				
				**B*6701**		675					
				**B*7301**		3347	2171				
				**B*7801**		6089	4597				
				**B*8101**		1352	729				
				**B*8201**		4367	3069				
				***B* alleles***	***0***	***48***	***44***				

**Table 5 vaccines-09-00680-t005:** Evidence showed that HLA-I polyreactive mAb TFL-006 bound to OCs (β2m-free αHLA HC) but not to CCs (β2m-associated HLA heavy chains or intact HLA molecules). TFL-006 bound only with LABScreen SABs (contained both open and CCs) but not with LIFECODES, which had only CCs, as established in previous reports [45,47].

TFL-006 (20 ug/mL)																																											
HLA-A	NC	PC	A*01:01	A*02:01	A*02:03	A*03:01	A*11:01	A*11:02	A*23:01	A*24:02	A*24:03	A*25:01	A*26:01	A*29:01	A*29:02	A*30:01	A*31:01	A*32:01	A*33:01	A*33:03	A*34:02	A*36:01	A*43:01	A*66:01	A*66:02	A*68:01	A*68:02	A*69:01	A*74:01	A*80:01													
	REGULAR LABSCreen BEADSET (Lot: LS1A04-lot 10) [*Closed Conformers admixed with Open Conformers*]																																										
	0	12	933	339	1018	193	4782	537	133	716	2516	194	2221	1017	778	1496	396	515	1038	554	1535	1353	2479	1886	1454	713	1185	3128	652	3132													
	LIFECODES BEADSET (Lot # 3005619) (Closed Conformers only)																																										
	0	0	0	0	0	0	0	0	0	0	0	0	0	0	0	0	0	0	0	0	0	0	0	0	0	0	0	0	0	0													
HLA-B	B*07:02	B*08:01	B*13:02	B*14:01	B*14:02	B*15:01	B*15:02	B*15:03	B*15:12	B*15:13	B*15:16	B*18:01	B*27:05	B*27:08	B*35:01	B*37:01	B*38:01	B*39:01	B*40:01	B*40:02	B*41:01	B*42:01	B*44:02	B*44:03	B*45:01	B*46:01	B*47:01	B*48:01	B*49:01	B*50:01	B*51:01	B*52:01	B*53:01	B*54:01	B*55:01	B*56:01	B*57:01	B*58:01	B*59:01	B*67:01	B*73:01	B*78:01	B*81:01
	REGULAR LABSCreen BEADSET (Lot: LS1A04-lot 10) [*Closed Conformers admixed with Open Conformers*]																																										
	862	1226	2514	7805	1831	335	1935	1822	770	3135	3076	3096	634	1659	6128	2650	2521	704	3429	2697	3739	347	3650	1829	1736	3572	2152	3262	1554	1799	2461	2146	5442	1662	2519	3662	2089	5268	3553	406	1423	2996	1525
	LIFECODES BEADSET (Lot # 3005619) (Closed Conformers only)																																										
	0	0	12	0	3	0	0	0	0	0	0	0	0	0	0	0	0	0	0	0	0	0	0	0	0	0	0	10	0	0	8	0	2	0	0	0	0	0	1	0	2	0	0
HLA-C	C*01:02	C*02:02	C*03:03	C*03:04	C*04:01	C*05:01	C*06:02	C*07:02	C*08:01	C*14:02	C*15:02	C*16:01	C*17:01																														
	REGULAR LABSCreen BEADSET (Lot: LS1A04-lot 10) [*Closted Conformers admixed with Open Conformers*]																																										
	4066	7446	2458	4504	3337	9124	5644	8702	6090	3937	4465	4648	8296																														
	LIFECODES BEADSET (Lot # 3005619) (Closed Conformers only)																																										
	1	20	0	0	8	24	4	97	1	0	4	0	10																														

**Table 6 vaccines-09-00680-t006:** HLA-I reactivity of different therapeutic preparations of IVIg.

Therapeutic Preparations of IVIg	Reactivity of Different HLA Class I Antigens					
	Classical HLA-Ia Alleles			Non-Classical HLA-Ib		
	A	B	Cw	E	F	G
IVIg (GamaSTAN, Talecris Biotherapeutics, Inc., Research Triangle Park, NC, USA)	31	50	16	*Positive*	*Positive*	*Positive*
IVIg (Octogam, Octapharma S.A. Argentina Poniente, Mexico, D.F.)	30	47	16	*Positive*	*Positive*	*Positive*
IVIg (Sandoglobulin, CSL Behring, Kankakee, IL, USA)	30	47	16	*Positive*	*Positive*	*Positive*
IVIg (GlobEx, Bangalore, India)	20	39	16	*Positive*	*Positive*	*Positive*
IVIg (IV-LFB-CNTs LFB Biomedicaments, Courtaboeuf Cedex, France)	31	50	16	*Positive*	*Positive*	*Positive*

**Table 7 vaccines-09-00680-t007:** Comparison of natural and functional characteristics of IVIg and HLA-I polyreactive mAbs.

**Source, Nature, and Functions**	**Intravenous Immunoglobulin (IVIg)**	**HLA-I Polyreactive mAbs, TFL-006 & TFL-007**
Manufacturer	Several pharmaceutical firms	Terasaki Foundation Laboratory, U.S. Patent No. 10,800.847;10/13/20
Source	Purified from pooled plasma of 10,000 blood donors	Immunized in mice with a heavy chain of HLA-E^R107^
Nature of antibody	Human, polyclonal IgG with trace levels of IgA	Murine purified monoclonal IgG
Subclass of IgG antibodies	IgG1, IgG2a, IgG3, IgG4	IgG2a
Purity	Contains soluble HLA antigens and other non-IgG proteins	100% purified protein of IgG2a [44,45]
	Cytokines, chemokines	
Antibody reactivity	CCs & OCs of HLA-A, HLA-B, HLA-Cw, HLA-E, HLA-F	OCs but not CCs of HLA-A, HLA-B, HLA-Cw, HLA-E, HLA-F, HLA-G
	HLA-G, HLA-DR, HLA-DQA/DQB, HLA-DPA/DPB	None
	Fc receptors: FcgI, FcgII, FcgIII, FcgIV (tested) [96]	FcgII (anticipated)
	Blood groups: A, B, Rh	No
	*Escherichia coli* bacterial antigens ranging from	No
	antigens by different preparations of IVIg	No
	Human albumin	No
	Phospholipids	No
Binding site	Binds to both closed and open conformers	Binds only to open conformers
Stabilizer	Many, including sucrose in some preparations	None
Protein concentration	Highly variable, from 2 to 12%	Protein concentration adjusted to requirement
CD4+ T-cell suppression	PHA- or cytokine-activated T cells by apoptosis and necrosis	PHA-activated T cells
CD8+ T-cell proliferation	PHA-activated T-cells	PHA-activated T cells
B-cell proliferation	May induce differentiation	None
Anti-HLA antibody suppression	PRA antibody reduction	Suppress production of anti-HLA-I and anti-HLA-II IgG
	Suppress selected HLA-II antibody production	
	Promote selected HLA-II antibody production	
Expansion of Tregs	Yes	Yes
Special application	Not applicable	To monitor the presence of open conformers admixed with
		closed conformers on the bead sets. (e.g., LABScreen vs
		LIFECODE bead sets used in monitoring HLA antibodies

## Data Availability

Data are available with the first author of each investigation [58,92]/./;[−00,847; 13 October 2020).

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
