# Peer review of "Therapeutic Potential of HLA-I Polyreactive mAbs Mimicking the HLA-I Polyreactivity and Immunoregulatory Functions of IVIg"

_vaccines, 2021, doi:10.3390/vaccines9060680_

Round 1

Reviewer 1 Report

It is a work of great interest, as well as the proposal for the implementation of techniques and molecules that allow us to identify different immunoregulatory functions.

Among the observations to the manuscript:

  • Line 60 (..),
  • Line 183 (forms. Of)
  • A grid with no data appears in the table below the same lines 353-355.
  • Figure 11 shows letters that are not from the figure or the image is manipulated and not mentioned in the footnote.
  • In addition to the quality of the images, which in some cases taken from quotes by the same author, could improve the resolution and mention that they are taken and modified or mention whether it is de novo or in original format.
  • Table 7 is not explicitly stated in the text: “Possibly due to polyclonality and mixing of various types of IgG Abs present, including Abs against all HLA class I loci and alleles as illustrated in Table and Figure 5A, and 5B” in line 143-144

From the background of the manuscript:

It does not mention the type of information search carried out, such as a search for convenience or an intentional. My suggestion is to explain the analysis of the experimental data performed, and the rationale for why only perform experiments for the hypotheses of immunomodulation and cell proliferation. As well as the ethical approval of the experimental analysis.

The advantages or disadvantages of other technologies compared to the proposed HLA mAbs, the methods and the validations of the technique are unclear.

Author Response

REPLY TO  REVIEWER # 1

It is a work of great interest, as well as the proposal for the implementation of techniques and molecules that allow us to identify different immunoregulatory functions.

We thank the reviewer for considering the review as “a work of great interest” and for opening the eyes of the authors to several lacunae and errors in the manuscript. We have recognized and corrected the errors as follows

Among the observations to the manuscript:

  • Line 60 (..),

Corrected

  • Line 183 (forms. Of)

Corrected

  • A grid with no data appears in the table below the same lines 353-355.

Corrected

  • Figure 11 shows letters that are not from the figure or the image is manipulated and not mentioned in the footnote.

Corrected. The font size of the letters in Y axis and the word Gamunex –C is added. This is a original figure.

  • In addition to the quality of the images, which in some cases taken from quotes by the same author, could improve the resolution and mention that they are taken and modified or mention whether it is de novo or in original format.

Carried out for each figure as suggested by the reviewer.

  • Table 7 is not explicitly stated in the text: “Possibly due to polyclonality and mixing of various types of IgG Abs present, including Abs against all HLA class I loci and alleles as illustrated in Table and Figure 5A, and 5B” in line 143-144

We specially thank the reviewer for bring this major error to our attention. We have now included it in the text.

From the background of the manuscript:

It does not mention the type of information search carried out, such as searching for convenience or an intentional. My suggestion is to explain the analysis of the experimental data performed, and the rationale for why only perform experiments for the hypotheses of immunomodulation and cell proliferation. As well as the ethical approval of the experimental analysis.

The advantages or disadvantages of other technologies compared to the proposed HLA mAbs, the methods and the validations of the technique are unclear.

We have taken efforts to revise the manuscript in several places to explain the analysis of the experimental data and the rationale as shown in red. However, we have cited our previous publication which explains the analysis of the experimental data performed and the rationale underling each each experiment. In this review, we have provide a condensed version.

We have also provided information about IRB approval and regarding animal use.

The method and validation are shown by comparing the results of Luminex SAB assay obtained with regular beads, iBeads and acid treated  denatured beads and explained in the text appropriately.

PLEASE SEE THE REVISED PREPRINT VERSION WITH ALL CORRECTIONS IN THE TEXT IN RED.

Reviewer 2 Report

vaccines-1221454

Mepur H. Ravindranath, Fatiha E. Hilali, Edward J. Filippone

Title: Therapeutic potential of HLA-I polyreactive mAbs mimicking the HLA-I polyreactivity and immunoregulatory functions of IVIg

This is a well constructed and written review of HLA class-I (HLA-I) polyreactive monoclonal antibodies. The authors obviously have spent a great deal of time and effort in putting together this review. There are some items that can improve the manuscript.

Organization: The review is organized into sections that are numerically categorized. What is missing would be a description of the different sections at the end of section #1. This would help the reader to understand what is coming in the review.  

Spelling: There are misspellings throughout the document that are too numerous to elaborate. The authors are encouraged to carefully go through the manuscript and make corrections.

Grammar: Some noticeable grammar corrections that are needed:

page 2, line 57 – "some" instead of "few"

page 26, line 366 – "involves" instead of "involve"

Other corrections may be needed and the authors should go through the document carefully.

Author Response

REPLY TO REVIEWER # 2 

This is a well constructed and written review of HLA class-I (HLA-I) polyreactive monoclonal antibodies. The authors obviously have spent a great deal of time and effort in putting together this review. There are some items that can improve the manuscript.

 We thank the reviewer for considering the review as “well constructed and written” and suggesting aspects that can improve the review.

Organization: The review is organized into sections that are numerically categorized. What is missing would be a description of the different sections at the end of section #1. This would help the reader to understand what is coming in the review.  

We havr now provided a description of the different sections at the end of the section # 1 as follows:

The objective of this review is to compare the nature and functional characteristics of the polyreactive anti-HLA-E mAbs (TFL-006 and TFL-007) with those of the commercial preparations of therapeutic IVIg as follows:

  • Determine if HLA-E shares antigenic amino acid sequences       (epitopes) common to all other HLA-I isoforms;
  • Document HLA-I polyreactivity of HLA-E mAbs;
  • Document HLA-I polyreactivity of the therapeutic preparations       of  IVIg;
  • Compare immunomodulation by IVIg with polyreactive mAbs;
  • Suppression of T-cell proliferation;
  • Suppression of antibody production by B cells;
  • Expansion of Foxp3+ Tregs.

Spelling: There are misspellings throughout the document that are too numerous to elaborate. The authors are encouraged to carefully go through the manuscript and make corrections.

We thank the reviewer for pointing out the spelling and grmmatical errors. We have now extensively revised and all corrections made are in RED. See the revised mansucript for the corrections made. 

Grammar: Some noticeable grammar corrections that are needed:

page 2, line 57 – "some" instead of "few" Corrected

page 26, line 366 – "involves" instead of "involve" Corrected.

Other corrections may be needed and the authors should go through the document carefully.

We observed several such errors and all of them to our best of knowledge were corrected.

PLEASE SEE THE REVISED PREPRINT VERSION WITH ALL CORRECTIONS IN THE TEXT IN RED.

Reviewer 3 Report

Ravindranath Mh et al. present a review on the therapeutic potential of HLA-class I reactive monoclonal antibodies comparing the functional properties to IVIG. The review is well written and covers lot of ground. I think the review could benefit from re-organization and a logical flow of information. The sections feel disjointed and leaves the reader confused.

I have the following comments

  1. The review can be shortened, there is lot of information that the average immunology student will already be familar with and doesnt need additional explanation (for e.g CFSE based cell proliferation).
  2. Figures are not that clear and crisp. May be suggest to include high resolution figures.
  3. Its unclear how HLA-I mabs play role in PHA induced activation of T-cells considering that mechanism of action is mainly by TCR crosslinking ? can the authors explain why
  4. The authors dont discuss any effect of HLA-I Mabs on NK cells ? 
  5. Are there any animal studies to do a direct comparison of IVIG vs HLA-I mabs ?

Author Response

REPLY TO REVIEWER # 3

Ravindranath Mh et al. present a review on the therapeutic potential of HLA-class I reactive monoclonal antibodies comparing the functional properties to IVIG. The review is well written and covers lot of ground. I think the review could benefit from re-organization and a logical flow of information. The sections feel disjointed and leaves the reader confused.

We thank the reviewer for considering the review well written and the comment on re-organization and a logical flow of information. Based on the reviewers recommendation,  we have extensively revised the manuscript, and the revisions are shown in red. In addition, to clarify the logical flow of information, we have introduced the following at the end of section # 1

The objective of this review is to compare the nature and functional characteristics of the polyreactive anti-HLA-E mAbs (TFL-006 and TFL-007) with those of the commercial preparations of therapeutic IVIg as follows:

  • Determine if HLA-E shares antigenic amino acid sequences                  (epitopes) common to all other HLA-I isoforms;
  • Document HLA-I polyreactivity of HLA-E mAbs;
  • Document HLA-I polyreactivity of the therapeutic      preparations of  IVIg;
  • Compare immunomodulation by IVIg with polyreactive      mAbs;
  • Suppression of T-cell proliferation;
  • Suppression of antibody production by B cells;
  • Expansion of Foxp3+ Tregs.

Then we proceed with different sections modified from that of the original version.

We have the following comments

  1. The review can be shortened, there is lot of information that the average immunology student will already be familiar with and doesn't need additional explanation (for e.g CFSE based cell proliferation).

First author and second author spend lot of time discussing the critique of the reviewer and taken effort to shorten the version and deleted aspects (such as CFSE) in the revised version.

  1. Figures are not that clear and crisp. May be suggest to include high resolution figures.

            We tried to improved the figures by increasing the pixel. To some extend we could      achieve it.

  1. Its unclear how HLA-I mabs play role in PHA induced activation of T-cells considering that mechanism of action is mainly by TCR crosslinking ? can the authors explain why

            The reviewer is correct that the TCR-crosslinking activates T-cells by Phorphorylating CD3. But Phosphorylation is not restricted to CD3 but also to other cell surface molecules overexpressed during activation, which includes HLA OCs.

On the advice of the reviewer, we have revised as follows:

The TCR crosslinking leads to expression and phosphorylation of cell surface molecules such as IL-Ra [74] and OCs of HLA-I [13 -22]. The binding of mAbs to the shared amino acid sequences or epitopes exposed on the OCs may dephosphorylate the tyrosyl and seryl residues on the elongated cytoplasmic tail of the HLA-I OCs [75 – 77].  This may simultaneously lead to dephosphorylation of CD3 and revert the PHA-activation of CD3 on T-cells (for further details, see in the legend for Figure 9).

  1. The authors dont discuss any effect of HLA-I Mabs on NK cells? 

We have purposely omitted NK cells inhibition, for we have shown earlier that it is MONOSPECIFIC AND NOT POLYREACTIVE mAbs are invovled in the inhibition process. This review is restricted to Polyreactive mAbs mimicking IVIg.

The reviewr is referred to our previous articles on this subject

  1. Ravindranath MH, Terasaki PI, Pham T, Jucaud V. The Monospecificity of Novel Anti-HLA-E Monoclonal Antibodies Enables Reliable Immunodiagnosis, Immunomodulation of HLA-E, and Upregulation of CD8+ T Lymphocytes. Monoclon Antib Immunodiagn    Immunother.        2015; 34(3): 135–53. DOI: 10.1089/mab.2014.0096. (NOT CITED IN THIS REVIEW)
  1. Ravindranath, M. H., and El Hilali, F. (2021) Monospecific and Polyreactive MAbs against Human Leukocyte Antigen-E: Diagnostic and Therapeutic Relevance, Chapter in “MAbs” 38p. Ed. Nima          Resaei, 2021, Intech Open. United Kingdom, ISBN: 978-1-83968-370-1.

  1. Are there any animal studies to do a direct comparison of IVIG vs HLA-I mabs ?

Research in progress. Too preliminary,

PLEASE SEE THE REVISED PREPRINT VERSION WITH ALL CORRECTIONS IN THE TEXT IN RED.